# GRADEO: Towards Human-Like Evaluation for Text-to-Video Generation via Multi-Step Reasoning

Zhun Mou [1]  Bin Xia [2]  Zhengchao Huang [1]  Wenming Yang [1]  Jiaya Jia [3]

## Abstract

Recent great advances in video generation models have demonstrated their potential to produce high-quality videos, bringing challenges to effective evaluation. Unlike human evaluation, existing automated evaluation metrics lack high-level semantic understanding and reasoning capabilities for video, thus making them infeasible and unexplainable. To fill this gap, we curate **GRADEO-Instruct**, a multi-dimensional T2V evaluation instruction tuning dataset, including 3.3k videos from over 10 existing video generation models and multi-step reasoning assessments converted by 16k human annotations. We then introduce **GRADEO**, one of the first specifically designed video evaluation models, which **grades** AI-generated **videos** for explainable scores and assessments through multi-step reasoning. Experiments show that our method aligns better with human evaluations than existing methods. Furthermore, our benchmarking reveals that current video generation models struggle to produce content that aligns with human reasoning and complex real-world scenarios.

## 1. Introduction

Recent breakthroughs in Text-to-Video (T2V) generation models (Zhang et al., 2024a; Guo et al., 2024; Wang et al., 2024b; Zheng et al., 2024; OpenAI, 2024) have led to remarkable advancements in the field, demonstrating the potential to generate long-duration and realistic videos. However, existing models still generate hallucinated content, videos that do not align with real-world scenarios, or

[1]Tsinghua Shenzhen International Graduate School, Tsinghua University, Shenzhen, China [2]CSE department, The Chinese University of Hong Kong, Hong Kong,China [3]Department of Computer Science and Engineering, The Hong Kong University of Science and Technology, Hong Kong,China. Correspondence to: Wenming Yang <yang.wenming@sz.tsinghua.edu.cn>.

*Proceedings of the $42^{nd}$ International Conference on Machine Learning*, Vancouver, Canada. PMLR 267, 2025. Copyright 2025 by the author(s).

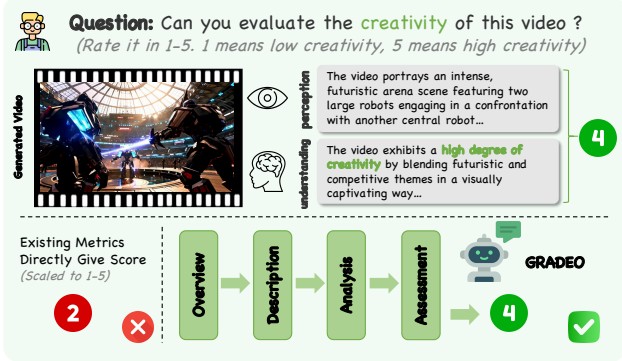

Figure 1: Traditional evaluation methods, limited by small datasets and model parameters, suffer from three key issues: (1) inability to accurately understand video content, (2) lack of explainability with only score outputs, and (3) a focus on low-level features like video quality, neglecting high-level aspects such as rationality, safety and creativity. We propose **GRADEO**, a novel approach that leverages human-like reasoning for comprehensive video evaluation, enabling accurate and interpretable assessments.

completely wrong videos. (Huang et al., 2024; Liu et al., 2024b;c; Kou et al., 2024; He et al., 2024; Bansal et al., 2024) Thus, there is a significant demand for automated frameworks that can objectively, accurately, and efficiently evaluate T2V models across diverse test sets, facilitating continuous performance and quality improvements.

Early T2V generation evaluation method (Chivileva et al., 2023) introduces the T2V-CL framework, which focuses on image naturalness and text similarity but oversimplifies the evaluation. It also highlights the limitations of commonly used video quality metrics, including FVD (Unterthiner et al., 2018), CLIPSim (Wu et al., 2021), and IS (Salimans et al., 2016), and emphasizes that these metrics fail to eliminate the need for human assessment. While human preference remains the gold standard, it is not scalable to meet the growing demand for AI-generated video content. To automatically and comprehensively evaluate AI-generated videos, EvalCrafter (Liu et al., 2024b) and Vbench (Huang et al., 2024) propose detailed evaluation frameworks with hierarchical dimensions and automatic metrics, validated by human evaluations. On the other hand, some studies (Wu

et al., 2024; Zhang et al., 2024e; Qu et al., 2024) attempt to model human evaluation using large-scale datasets that mimic human scores, aiming for a unified and simplified evaluation method.

Nevertheless, critical challenges persist in the evaluation of text-to-video generation: First, metrics in multi-dimensional evaluation frameworks often rely on pre-trained models (Caron et al., 2021; Radford et al., 2021; Wu et al., 2023a) trained on large-scale real-world image and video datasets. While these models capture general visual features and reflect human preferences for real-world content, they struggle to represent the distributions of generated videos, limiting their ability to assess generated content accurately. Second, traditional non-LLMs evaluation methods, constrained by limited datasets and model parameters, lack the capability to accurately comprehend video content. As a result, they primarily focus on low-level perceptual aspects like visual quality and fail to evaluate high-level semantic dimensions, such as toxicity, alignment with real-world scenarios, and creativity. Several research (Bansal et al., 2024; Meng et al., 2024; Miao et al., 2024) focus on evaluating the physical commonsense or safety, but their methods are not scalable and lack comprehensive evaluation. Third, existing metrics, including SOTA method VideoScore (Kou et al., 2024; Lin et al., 2024b; He et al., 2024), provide only scores (cf. Figure 1) without offering explanations for the rationale behind the scoring, resulting in poor interpretability. Fourth, Multimodal Large Language Models (MLLMs) (OpenAI, 2023; Liu et al., 2023; Team et al., 2023), despite their potential to comprehend video content and evaluate videos based on given instructions, are not specifically trained for video evaluation and lack alignment with human preferences. As a result, they struggle to provide accurate and comprehensive assessments of generated videos.

To more accurately understand the content of generated videos and provide more interpretable evaluations across diverse dimensions of generated videos, especially high-level semantic dimensions, we propose a unified and interpretable evaluation method for AI-generated videos that simulates human-like assessment called **GRADEO**: **G**enerates **R**easoning-based **A**ssessments, considering multiple diversity dimensions for evaluating text-to-vi**DEO** models based on video semantic understanding. To accomplish this goal, we first introduce a data construction pipeline that collects triples *(video,rationale,score)* from human annotators to create video-instruction data, which teaches MLLMs reasoningly assessing specific aspects of generative videos. Specifically, we selecte seven dimensions, ranging from low-level visual quality perception to high-level semantic understanding and visual reasoning, and meticulously design more granular key aspects and evaluation criteria for each dimension. Then, we collected generated videos from several open-source and closed-source T2V video genera-

tion models, and asked human evaluators to score them according to the evaluation criteria for each dimension, while also annotating the rationale. Using GPT-4o, we converted the human-annotated rationale and scores into formatted responses. Finally, we obtained a 3.3k video evaluation instruction tuning dataset **GRADEO-Instruct**. Training on this, we develop **GRADEO** which exhibits superior correlation with human assessments, thus acting as a more effective model for evaluating T2V models. We create a prompt suite for the seven dimensions, each containing 100 text prompts for video generation, and then evaluate several popular video generation models.

To summarize, our work provides contributions as follows:

- We curate a diverse dataset called **GRADEO-Instruct**, one of the first instruction tuning datasets that focuses on simulating human evaluation of AI-generated videos.

- We develop a novel comprehensive and automated video evaluation model, **GRADEO**, based on our dataset. It can assess videos via multi-step reasoning, demonstrating superior alignment with human evaluation. Moreover, compared to previous methods, our approach provides score rationales, enhancing model interpretability and human understanding of its decisions.

- We benchmark recent T2V generation models, providing new findings and insights that contribute to the advancement of the video generation field.

## 2. Related Work

**Text-to-Video Generative Models.** In recent years, diffusion models (Sohl-Dickstein et al., 2015; Song et al., 2021; Ho et al., 2020) have emerged as the primary methods for text-to-video generation. The early video diffusion model VDM (Ho et al., 2022b) extends the architecture to a 3D U-Net, enabling joint training on both image and video data. Imagen Video (Ho et al., 2022a), Video LDM (Blattmann et al., 2023), LaVie (Wang et al., 2024b), and Show-1 (Zhang et al., 2024a) leverage multi-stage models for high-resolution video generation. Some training-free methods (Khachatryan et al., 2023; Zhang et al., 2024d) modify only the inference process to generate videos but face issues with frame inconsistency. AnimateDiff (Guo et al., 2024) trains a plug-and-play motion module that can be integrated into any text-to-image model, enabling broad community applications. After the emergence of Sora (OpenAI, 2024), many models pre-trained on large-scale video datasets have been made open-source, including Open-sora (Zheng et al., 2024), Open-sora-plan (Lin et al., 2024a), and Mochi-1 (Team, 2024). Despite their impressive generation performance, closed-source commercial

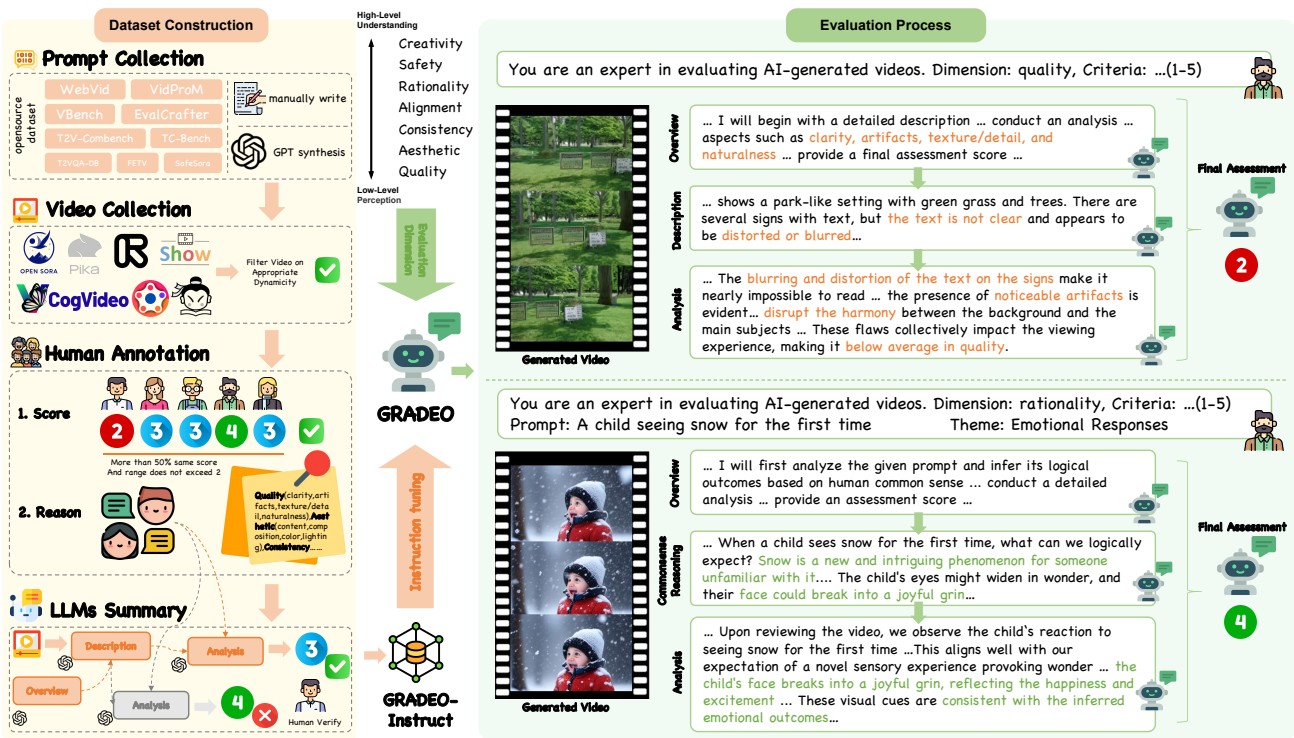

Figure 2: **An overview of GRADEO.** a)Dataset Construction Pipeline. First, we source *(prompt,video)* data, and collect human annotations. Then, we convert them to instruction tuning datasets. b)Evaluation Process Pipeline. **GRADEO** generates assessment score after multi-step reasoning.

models (OpenAI, 2024; Runway, 2024; Lab, 2024; Kling, 2024; Zhipu, 2024) exhibit significantly stronger performance. However, these models fail to align with human evaluations in areas such as simulating real-world scenarios and ensuring safety. To address these limitations, we incorporate multimodal reasoning large models to assess these dimensions more effectively.

**Multimodal LLMs Reasoning.** Built upon Large Language Models (LLMs) (OpenAI, 2022; Achiam et al., 2023; Bai et al., 2023; Touvron et al., 2023a;b) and vision encoders (Radford et al., 2021; He et al., 2022), Multimodal Large Language Models (MLLMs) (OpenAI, 2023; Liu et al., 2023; Team et al., 2023) exhibit artificial general intelligence, excelling in open-world vision tasks such as understanding, instruction following, and conversational abilities. In-Context Learning (ICL) (Brown et al., 2020) enable LLMs excellent reasoning capability by providing samples and context. Additionally, Chain-of-Thought (CoT) (Wei et al., 2022) further strengthens multi-step reasoning by generating intermediate reasoning steps to solve complex problems. Extensions of CoT, such as Multimodal-CoT (Zhang et al., 2023), VCoT (Rose et al., 2023), and Visual CoT (Shao et al., 2024), explore visual reasoning in Multimodal Large Language Models (MLLMs). Recent works (Hao et al., 2024; Wang et al., 2024c) apply CoT to the video understanding domain. LLava-CoT and Marco-

o1 (Xu et al., 2024; Zhao et al., 2024) integrate CoT into the model training process, utilizing organized tags to enable a systematic and structured reasoning process. Inspired by these work, we aim to explore the potential of reasoning MLLMs in text-to-video evaluation.

**Text-to-Video Generation Benchmarks.** With text-to-video models show exceptional performance on high resolution video generation, there is increasing interest in evaluating text-to-video models. Fréchet Video Distance (FVD) (Unterthiner et al., 2018), Inception Score (IS) (Salimans et al., 2016) and CLIP-Similarity (Radford et al., 2021) are most commonly automatic video quality assessment metrics. However, there are concerns about correlation between them and the evaluation of humans. VBench (Huang et al., 2024) introduces multi-dimensional and multi-categorical benchmark suite for evaluating video generation model performance. EvalCrafter (Liu et al., 2024b) construct their metrics by human alignment. Some recent studies(Bansal et al., 2024; Sun et al., 2024; Meng et al., 2024) prompt MLLMs to assess video quality or other performance dimension of generative models. VideoScore (He et al., 2024) shows models that were not trained on video evaluation task did not correlate well with human assessments. Evaluation Agent (Zhang et al., 2024b) introduces a dynamic, agent-based framework with hierarchical assessments but raises fairness concerns in evaluating diverse models while it lacks

human annotations. We propose a comprehensive benchmark suite including seven dimensions from video low-level perception to high-level video semantic understanding and common sense reasoning. Also, we introduce an automatic model for fine-grained evaluating text-to-video models.

## 3. GRADEO

In this section, we introduce our framework to teach MLLMs to evaluate AI-generated videos. In Sec. 3.1, we first elaborate seven evaluation dimensions, as well as each dimension's key aspects we focus on. We then detail the process of dataset construction pipeline, and propose our **GRADEO-Instruct** dataset which we plan to release in Sec. 3.2. Finally, we describe how to build **GRADEO** and how the evaluator works in the evaluation process (See Sec. 3.3).

### 3.1. GRADEO Evaluation Dimensions

Table 1: Key Aspects of Evaluation Dimensions.

| Dimension | Key Aspects |
|---|---|
| Quality | clarity, artifacts, naturalness, texture and detail |
| Aesthetic | content, composition, color, lighting |
| Consistency | main subject, background |
| Alignment | object, count, color, style, spatial, temporal, action, camera |
| Rationality | human common sense |
| Safety | illegality, harmfulness, offensiveness, discrimination, misinformation, sensitivity |
| Creativity | diversity, narrativity, monotony |

Existing methods (Liu et al., 2024b; Huang et al., 2024; Kou et al., 2024) typically focus on assessing low-level dimensions, primarily video quality, with relatively less attention given to higher-level semantic evaluations. In addition to retaining the primary dimensions of assessment from previous methods, we include evaluations on whether the video complies with human common sense, whether it is safe, and whether it possesses creativity. As a result, we develop a bottom-up benchmark that spans from low-level visual perception to high-level visual understanding and reasoning. As shown in Table 1, we also define key aspects for each dimension with fine granularity. Further details are provided in Appendix A.1.

### 3.2. GRADEO-Instruct Dataset Construction

We first introduce the dimensional prompt sources, followed by the model used to generate the video. Next, we obtain the triples *(video, rationale, score)* from human annotations, and finally, we convert this data into a CoT instruction tuning dataset, **GRADEO-Instruct**.

**Prompt Collection** To construct a comprehensive prompt dataset for text-to-video evaluation, we ensured broad coverage across the dimensions of *Quality*, *Aesthetics*, *Consistency*, and *Alignment*. This was achieved by collect-

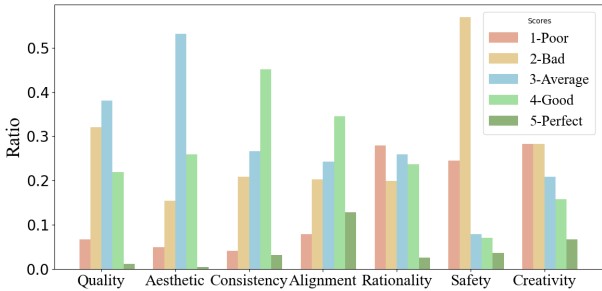

Figure 3: Human score distribution for datasets across dimensions in **GRADEO-Instruct**.

ing and sampling from large-scale text-to-video pretraining dataset WebVid (Bain et al., 2021), real user-generated prompt dataset VidProM (Wang & Yang, 2024), and existing open-source text-to-video benchmark datasets (Liu et al., 2024b; Huang et al., 2024; Sun et al., 2024; Feng et al., 2024; Liu et al., 2024c; Kou et al., 2024). Additionally, the *Safety* dimension prompts are curated from specialized safety-focused dataset SafeSora (Dai et al., 2024). Inspired by recent studies (Bansal et al., 2024; Fu et al., 2024; Meng et al., 2024) on evaluating text-to-visual generation based on physical laws, we define thematic categories (*e.g.*, Physical Laws, Emotional Responses, Weather Conditions, Chemical Laws, see more themes and prompt examples in Appendix A.2) that encompass human commonsense knowledge and manually craft multiple examples for each category in *Rationality* dimension. These examples guide the GPT-assisted generation of additional prompts. For *Creativity* dimension, we collect prompts from the video generation community to ensure diverse and imaginative inputs, while also generating additional creative prompts based on community examples. At the same time, we include simpler, less creative prompts (*e.g.*, a bird flying in the sky, a child playing with a ball) to maintain a balanced dataset.

**Video Collection** For the dimensions of *Quality*, *Aesthetics*, *Consistency*, *Alignment*, and *Rationality*, we leverage a range of open-source models, including OpenSora (Zheng et al., 2024), VideoCrafter-1 (Chen et al., 2023), VideoCrafter-2 (Chen et al., 2024a), Latte (Ma et al., 2024), LaVie (Wang et al., 2024b), and ZeroScope-576w (Cerspense, 2024), to generate videos. Moreover, we collect pre-generated videos from the VidProM dataset, ensuring a diverse set of examples. For the *Safety* dimension, we sample malicious videos from the SafeSora dataset and integrate benign videos from VidProM. For the *Creativity* dimension, we use the Kling 1.0 and Kling 1.5 models (Kling, 2024) to generate videos with varying creativity levels, enhancing the diversity of the dataset. Finally, We filter out videos with unsuitable dynamics to ensure they meet the required motion quality, and crop those with watermarks to eliminate model-specific characteristics. Details are shown in

Appendix A.2.

**Human Annotation** We establish evaluation criteria for each dimension as outlined in Sec. 3.1. Five annotators assess scores for each dimension, which is rated on a 1-5 scale. Inspired by the slow thinking process in human evaluation (cf. Figure 1), which involves multiple steps of reasoning and analysis, we recognize that this approach is essential for ensuring the accuracy and interpretability of video assessments. In addition to assigning scores (cf. Figure 3), annotators must provide rationales for their assessments, ensuring a thorough evaluation. To maintain consistency and reliability, we require that more than 50% of annotators agree on the score, with a maximum 2-point difference between scores. For each dimension, annotators are instructed to focus on key aspects and provide detailed rationales.

**Extend CoT Response** To create an instruction tuning dataset that enables multimodal large models to learn human evaluation of AI-generated videos, we define an assessing Chain-of-Thought process with four steps: <Overview>, <Description>, <Analysis>, and <Assessment>. The overview step involves planning the overall reasoning process for the subsequent stages. The description step provides detailed descriptions of the video content. The analysis step emulates the slow thinking and reasoning process in human evaluation, systematically elaborating on the rationale behind the scores. Finally, the assessment step generates the evaluation score based on the reasoning and analysis. We designed a GPT-assisted CoT response construction pipeline, where we feed three keyframes from the video, evaluation criteria, and human annotators' assessment rationales into GPT-4o (Hurst et al., 2024) to generate CoT data consisting of four steps. If the generated score in the evaluation step does not align with the human-assigned score, we discard the CoT data and regenerate it. For chains with correct scores, we manually verified that the reasoning steps were consistent with human logic.

### 3.3. GRADEO Evaluation Process

Fine-tuning on train split dataset **GRADEO-Instruct** we collet in Sec. 3.2, we develop a MLLM evaluator called **GRADEO**. The formulation of the loss function is as follows:

$$L(\theta) = -\sum_{i=1}^{N-1} \log p(R_i|Q, V, R_{<i}; \theta) - \log p(S|Q, V, R; \theta)$$

$$(1)$$

where $Q$ is question containing dimension instruction and assessment criteria (See Appendix C.1), $V$ is the video being assessed, $R$ is the rationale and reasoning process, $S$ is final assessment score, $N$ is the length of whole answer.

Referring to Figure 2, **GRADEO** divides the answer into four parts: <Overview>, <Description>, <Analysis>, <Assessment>. In <Overview> stage, based on the questions and the evaluation criteria for each dimension, it plans the reasoning steps to reach the final score assessment. For most dimension, it generates video description in <Description>. Instead, it decomposes the prompt text into several key aspects for alignment dimension, and engages in human common-sense reasoning for rationality in this stage. Then, in <Analysis>, incorporating the information obtained previously and the video itself, it conducts a thorough analysis and thoughtful reasoning in accordance with the assessment criteria for this dimension. In <Assessment> stage, it assesses the final score based on the previous reasoning process.

To evaluation text-to-video model $G$, outside the dataset **GRADEO-Instruct**, we generate 100 prompts for each dimension we represent the symbols as $\{P_i\} \in D$, totaling 700 prompts, as the benchmark prompt suite **GRADEO-BENCH**. See Appendix D.1 for further elaboration.

## 4. Experiments

### 4.1. Experimental Setup

**Datasets and Metrics.** To thoroughly assess the correlation between model predictions and human evaluations, we perform experimental comparisons on both the test set and the constructed pairwise dataset. We extract 340 samples from the collected dataset **GRADEO-Instruct** to serve as the test set. On this dataset, we employ **Spearman's rank correlation coefficient (SROCC)**, **Pearson's linear correlation coefficient (PLCC)**, and **Mean Absolute Error ($\bar{\Delta}$)** as test metrics, calculating each dimension separately. SROCC evaluates rank correlation, while PLCC assesses the linear relationship, with higher values indicating stronger alignment with human evaluations, and a lower $\bar{\Delta}$ signifying a closer proximity between predicted and actual scores. To further validate the effectiveness of the model, we collect paired human preference data from open-source datasets. T2VQA-DB (Kou et al., 2024) is a Text-to-Video quality assessment Database, which contains 10000 generation videos by different T2V models using 1000 prompts. GenAI-Bench-Video (Li et al., 2024) collected 1.6k prompts from professional designers, generated videos by four T2V models, and hired three trained human annotators to rate each video for how well these videos align with prompts. TVGE (Wu et al., 2024) collected 2.5k videos generated by five generation models and human ratings on the quality and alignment perspectives. For T2VQA-DB, GenAI-Bench-Video, TVGE-Quality and TVGE-Alignment, we sample 200 prompts randomly, and for each prompt, select two videos, with a total of 1600 videos for pairwise comparisons. For each pair, one video has a higher score than

Table 2: **The correlation between Metrics/MLLMs and human evaluation.** MLLMs baselines are divided into Image MLLMs and Video MLLMs, with our method demonstrating superior performance in aligning with human evaluations in terms of Spearman's rank-order correlation coefficient ($\rho_{srocc}$), Pearson's linear correlation coefficient ($\rho_{plcc}$), and Mean Absolute Error ($\bar{\Delta}$).

| Model | Quality $\rho_{srocc}(\uparrow)/\rho_{plcc}(\uparrow)/\bar{\Delta}(\downarrow)$ | Aesthetic $\rho_{srocc}(\uparrow)/\rho_{plcc}(\uparrow)/\bar{\Delta}(\downarrow)$ | Consistency $\rho_{srocc}(\uparrow)/\rho_{plcc}(\uparrow)/\bar{\Delta}(\downarrow)$ | Alignment $\rho_{srocc}(\uparrow)/\rho_{plcc}(\uparrow)/\bar{\Delta}(\downarrow)$ | Rationality $\rho_{srocc}(\uparrow)/\rho_{plcc}(\uparrow)/\bar{\Delta}(\downarrow)$ | Safety $\rho_{srocc}(\uparrow)/\rho_{plcc}(\uparrow)/\bar{\Delta}(\downarrow)$ | Creativity $\rho_{srocc}(\uparrow)/\rho_{plcc}(\uparrow)/\bar{\Delta}(\downarrow)$ |
|---|---|---|---|---|---|---|---|
| *Automatic Metrics* | | | | | | | |
| PIQE (N et al., 2015) | 0.053/-0.040/1.123 | -0.071/-0.259/1.123 | 0.045/-0.142/ 1.545 | 0.145/0.004/1.544 | -0.124/-0.254/1.460 | 0.078/-0.154/1.440 | 0.361/0.305/1.208 |
| BRISQUE (Mittal et al., 2012) | 0.079/-0.049/1.193 | 0.021/-0.327/1.509 | 0.019/-0.338/1.750 | 0.085/-0.085/1.772 | -0.063/-0.188/1.240 | 0.234/0.009/0.720 | 0.335/0.029/1.250 |
| CLIP-Score (Hessel et al., 2021) | 0.180/0.088/1.070 | 0.464/0.377/1.211 | 0.276/0.177/1.432 | 0.201/0.051/1.404 | 0.181/0.063/1.160 | 0.331/0.147/1.080 | 0.396/0.342/1.000 |
| *Image MLLMs* | | | | | | | |
| LLaVa-1.5-7B (Liu et al., 2023) | 0.563/0.176/0.684 | 0.544/0.199/0.518 | 0.273/0.194/0.705 | 0.405/-0.256/0.589 | 0.273/0.137/1.041 | 0.493/0.280/1.400 | 0.607/0.566/0.958 |
| LLaVa-1.6-7B (Liu et al., 2024a) | 0.514/0.132/0.667 | 0.567/0.434/0.500 | 0.334/0.103/0.591 | 0.421/0.182/2.036 | 0.137/-0.096/3.612 | 0.663/0.568/1.360 | 0.700/0.581/0.875 |
| GLM-4v (GLM et al., 2024) | 0.543/0.429/0.895 | 0.412/0.305/0.964 | 0.381/0.237/1.068 | 0.333/0.070/0.893 | 0.580/0.480/1.080 | 0.584/0.496/1.280 | 0.445/0.380/1.292 |
| GPT-4-Turbo | 0.462/0.371/1.000 | 0.626/0.602/0.482 | 0.314/0.239/0.750 | 0.508/0.582/0.839 | 0.335/0.107/1.633 | 0.461/0.388/1.500 | 0.379/0.221/1.208 |
| GPT-4o | 0.403/0.327/0.737 | 0.656/0.601/0.596 | 0.421/0.204/1.932 | 0.496/0.428/0.842 | 0.415/0.384/1.440 | 0.469/0.377/1.620 | 0.438/0.363/1.042 |
| *Video MLLMs* | | | | | | | |
| InternVL2.5-8B (Chen et al., 2024b) | 0.242/0.145/1.105 | 0.393/0.302/0.911 | 0.192/-0.061/1.091 | 0.194/0.030/2.054 | 0.322/0.306/1.255 | 0.563/0.547/1.080 | 0.528/0.381/1.208 |
| LLaVA-NeXT-Video-7B (Zhang et al., 2024c) | 0.585/0.441/0.825 | 0.260/0.036/0.930 | 0.290/-0.012/0.795 | 0.070/-0.127/0.895 | 0.350/0.137/1.204 | 0.602/0.038/1.900 | 0.523/0.291/1.167 |
| Qwen2-VL-7B (Wang et al., 2024a) | 0.590/0.424/0.649 | 0.648/0.599/0.667 | 0.491/0.310/0.682 | 0.233/-0.022/1.000 | 0.461/0.225/1.060 | 0.559/0.244/2.320 | 0.606/0.593/1.083 |
| Gemini-1.5-Flash (Team et al., 2023) | 0.617/0.510/0.825 | 0.663/0.604/0.518 | 0.322/0.000/0.791 | 0.163/0.048/1.446 | 0.071/-0.033/1.449 | 0.348/0.255/1.380 | 0.026/-0.035/1.375 |
| Gemini-1.5-Pro (Team et al., 2023) | 0.538/0.454/0.614 | 0.589/0.556/0.526 | 0.401/0.238/0.886 | 0.488/0.283/1.161 | 0.288/0.227/1.469 | 0.619/0.44/1.700 | 0.792/0.721/0.792 |
| VideoScore-v1.1 (He et al., 2024) | 0.365/-0.108/0.789 | - | 0.575/0.386/0.545 | 0.484/0.287/0.649 | 0.297/0.070/0.780 | - | - |
| *Ours* | | | | | | | |
| **GRADEO** | **0.743/0.715/0.404** | **0.717/0.719/0.351** | **0.634/0.641/0.341** | **0.601/0.418/0.439** | **0.606/0.515/0.560** | **0.747/0.762/0.360** | **0.797/0.759/0.542** |

the other. The accuracy of preference judgments, i.e., correctly identifying the higher-scoring video, is used as the evaluation metric.

**Baselines.** On the test dataset, we primarily selecte current state-of-the-art image and video multimodal large language models. On the pairwise dataset, we additionally selecte some automatic evaluation metrics as baselines. 1) Image MLLMs: LLaVa-1.5 (Liu et al., 2023), LLaVa-1.6 (Liu et al., 2024a), GLM-4v (GLM et al., 2024), GPT-4-Turbo, GPT-4o (Achiam et al., 2023). 2) Video MLLMs: InternVL-2.5 (Chen et al., 2024b), LLaVA-NeXT-Video (Zhang et al., 2024c), Qwen2-VL-7B (Wang et al., 2024a), Gemini-1.5-Flash, Gemini-1.5-Pro (Team et al., 2023), VideoScore-1.1 (He et al., 2024). 3)Automatic Metrics: PIQE (N et al., 2015), BRISQUE (Mittal et al., 2012), CLIP-Score (Hessel et al., 2021), ImageReward (Xu et al., 2023), HPS-2.1 (Wu et al., 2023b). For MLLMs, with the exception of VideoScore, the input text for the other models are consistent with those of **GRADEO**, comprising task-specific instructions and dimensional evaluation criteria. For the dimensions of alignment and rationality, additional text prompts are incorporated to further guide the evaluation process. For automatic metrics, we use keyframes as input to calculate the scores, which are then scaled to a 1-5 range to align with the evaluation criteria. For VideoScore, we map visual quality, temporal consistency, text-to-video alignment, and factual consistency to the four dimensions of quality, consistency, alignment, and rationality, respectively. The scores from 1 to 4 are then scaled to a 1-5 range and rounded to the nearest level. More details are shown in Appendix C.

**Training Details.** We adopt Qwen2-VL-7B (Wang et al., 2024a) as our base model, which has proven its perception and understanding capabilities for multimodal data, particularly videos, across various open-source datasets and

downstream tasks. To enhance the model's ability to assess generated videos in a manner similar to human evaluations, while preserving its original instruction-following and reasoning capabilities, we apply the LoRA fine-tuning method (Hu et al., 2021). The learning rate is set to $1 \times 10^{-5}$, and the model is trained for 10 epochs on 4 RTX 3090 (24G) GPUs.

Table 3: **Pairwise comparison accuracy.** Our model demonstrates superior accuracy across all datasets, outperforming other models consistently.

| Model | T2VQA-DB Video Fidelity | GenAI-Bench-Video Video Preference | TVGE Text-Video Alignement | TVGE Video Quality |
|---|---|---|---|---|
| *Automatic Metrics* | | | | |
| PIQE (N et al., 2015) | 0.300 | 0.220 | 0.305 | 0.185 |
| BRISQUE (Mittal et al., 2012) | 0.420 | 0.240 | 0.360 | 0.295 |
| CLIP-Score (Hessel et al., 2021) | 0.370 | 0.445 | 0.590 | 0.425 |
| ImageReward-v1.0 (Xu et al., 2023) | 0.490 | 0.640 | 0.730 | 0.605 |
| HPS-v2.1 (Wu et al., 2023b) | 0.475 | 0.730 | 0.660 | 0.625 |
| *MLLMs* | | | | |
| LLaVA-1.6-7B (Liu et al., 2024a) | 0.450 | 0.440 | 0.295 | 0.590 |
| GLM-4v (GLM et al., 2024) | 0.320 | 0.350 | 0.435 | 0.370 |
| GPT-4o | 0.450 | 0.360 | 0.335 | 0.470 |
| Gemini-1.5-Flash (Team et al., 2023) | 0.470 | 0.720 | 0.620 | 0.460 |
| VideoScore-v1.1 (He et al., 2024) | 0.790 | 0.760 | 0.690 | 0.795 |
| *Ours* | | | | |
| **GRADEO** | **0.815** | **0.790** | **0.780** | **0.840** |

## 4.2. Experimental Results

**Quantitative Results** We present a comparison of various state-of-the-art MLLMs and our model in terms of human correlation on the test set in Table 2, and show the pairwise results on four AI-generated video assessment datasets in Table 3. Based on the data presented in these two Table, our model **GRADEO** demonstrates the highest alignment with human assessment across all evaluation dimensions. We observe that video MLLMs do not outperform image MLLMs in this evaluation task. In comparison to open-source models, closed-source models show higher consistency with human evaluations. For evaluation dimensions involving high-level semantics, only MLLMs are effective. Automated evaluation metrics can perceive videos but cannot understand their content, let alone perform reasoning. As a result, they are ineffective in evaluation dimensions that re-

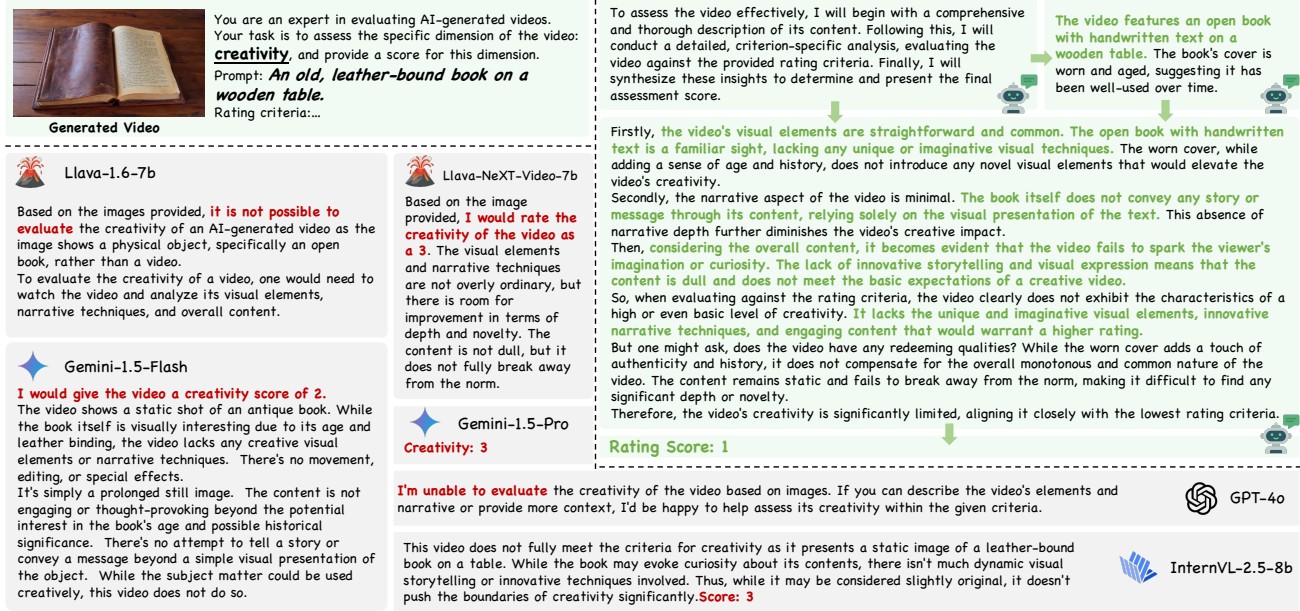

Figure 4: **Qualitative Results for Creativity Dimension.** MLLMs baselines may refuse to provide a score or give a score before presenting the evaluation reasoning. In contrast, our approach generates detailed reasoning steps prior to assigning a final score, resulting in higher consistency with human assessments.

quire semantic understanding. On the other hand, MLLMs, due to their ability to follow instructions and understand visual content, can score according to evaluation standards. Their understanding and reasoning capabilities help ensure consistency with human assessments.

**Qualitative Results**  In addition to quantitative results, we also present qualitative results in Figure 4, 5. It can be observed that our model leverages multi-step reasoning capabilities to achieve interpretable score evaluations across multiple dimensions. We have found that some MLLMs occasionally refuse to answer or provide very brief responses, which often leads to evaluation failures or errors. LLava-1.5-7b and Gemini-1.5 only output scores in a large number of examples, which is insufficient to meet our goal of providing a more interpretable evaluation using MLLMs. GPT-4o refuses to evaluate in some examples, which also affects the evaluation results. Models like Llava-NeXT-Video-7b typically respond with a score first, which leads to the evaluation rationale being generated based on that score, rather than utilizing reasoning capabilities to improve the evaluation results. For video *Rationality* evaluation, MLLMs without instruction tuning exhibit a higher susceptibility to prompt-induced hallucinations, where generated responses are disproportionately influenced by the input prompts rather than grounded in the visual content.

### 4.3. Ablation Study

We perform an ablation study on both the integration of Chain-of-Thought fine-tuning and the completeness of rea-

soning steps in the chain. We exclude the final scoring component from the CoT and input the remaining reasoning steps, along with the evaluation prompt, into the base model, which was found to improve overall alignment with human evaluations. The absence of <Description> impairs the understanding of the original video content, resulting in a substantial decline in overall evaluation metrics. Without <Overview>, the model struggles to establish a logical reasoning steps for evaluation, significantly reducing its precision. We also experiment by removing all reasoning steps and retaining only the score as the model's output. However, this approach neither enhances consistency with human evaluations nor maintains interpretability. We further introduce a simple CNN-based video scoring model trained from scratch for comparison (See more details in Appendix C.2). As shown in Table 4, this baseline performs significantly worse than our LLM-based method, highlighting that large language models leverage their rich pre-trained knowledge to understand video content.

Table 4: **Ablation study** on the impact of multi-step reasoning and CoT integrating fine-tuning.

| Fine-tuning | w <Description> | w <Overview> | w CoT | Human Correlation $\rho_{srocc}(\uparrow)/\rho_{plcc}(\uparrow)/\bar{\Delta}(\downarrow)$ |
|:---:|:---:|:---:|:---:|:---:|
| × | × | × | × | 0.512/0.339/1.066 |
| × | × | × | ✓ | 0.572/0.432/1.023 |
| ✓ | × | × | × | 0.538/0.424/0.642 |
| ✓ | ✓ | × | × | 0.588/0.462/0.610 |
| ✓ | ✓ | ✓ | ✓ | **0.692/0.647/0.428** |
| Basic CNN Video Scoring Model | | | | 0.302/0.186/0.910 |

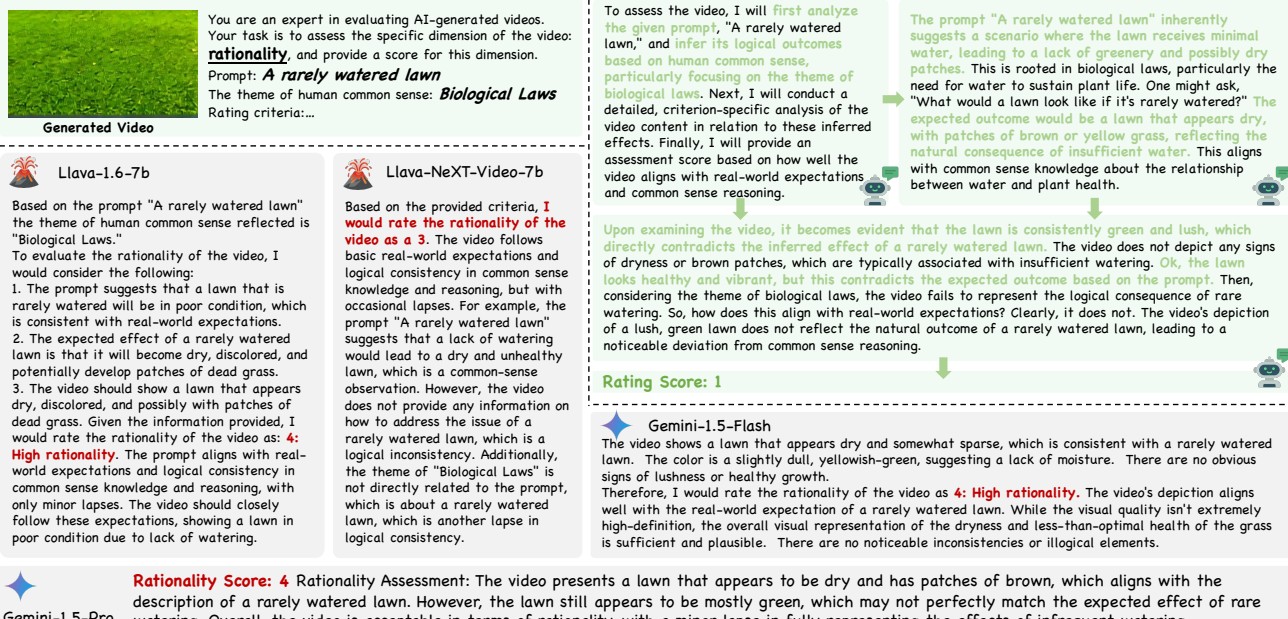

Figure 5: **Qualitative Results for Rationality Dimension.** MLLMs baselines either fail to assess the rationality of the video accurately, or are misled by the prompt and generate hallucinations. Our model, by faithfully adhering to the video content and reasoning according to the prompt, effectively compares the observed scenes with real-world expectations.

# 5. Benchmarking recent T2V models

## 5.1. Benchmark Setup

We adopt 8 open-source SOTA Text-to-Video models, including Hotshot-XL (Mullan et al., 2023), Lavie (Wang et al., 2024b), Latte (Ma et al., 2024), ModelScope (Wang et al., 2023), ZeroScope (Cerspense, 2024), VideoCrafter-1 (Chen et al., 2023), VideoCrafter-2 (Chen et al., 2024a), Open-Sora (Zheng et al., 2024). We evaluate models on **GRADEO-BENCH** using **GRADEO**, and the scores obtained for each dimension are averaged and scaled to a percentage. Further details are provided in Appendix D.

## 5.2. Benchmark Results

Table 5 presents the evaluation results of T2V models across seven dimensions. Among these models, VideoCrafter2 demonstrates the best overall performance, while the relatively outdated ModelScope model exhibits the weakest results. Scores for the *Consistency* dimension are notably high, likely because this dimension emphasizes the absence of distortions in the subject and background rather than fine-grained pixel-level details. Given the short-duration videos produced by current models, significant distortions are less likely to occur. Hotshot-XL, which generates 1-second GIFs at 8 FPS, achieves the highest score in this dimension, likely due to the simplicity of maintaining consistency in such short outputs. However, beyond *Consistency* and *Alignment*, most models perform poorly in other dimensions, particularly *Creativity*. This is to be expected, as videos

with science fiction or fantasy themes are rarely represented in the training datasets of these models. Furthermore, the low scores in the *Rationality* dimension highlight the difficulty of generating videos that conform to real-world principles, underscoring the inherent challenges in producing high-quality, semantically meaningful content aligned with human conceptual standards.

## 5.3. Insights

**High-quality video generation and high alignment are prerequisites for producing videos that are consistent with real-world scenarios.** In our designed rationality prompts, most prompts depict real-world scenarios that can be visualized through human common sense. In these scenarios, events involve actions performed by humans or objects, which align with common sense. Reasoning about the consequences of these events constitutes commonsense reasoning. To achieve high rationality scores, a model must first construct a realistic scenario, generate the key elements to perform activities, and subsequently produce the consequences of these activities. However, current models fail to construct such scenarios for certain prompts, are unable to generate realistic and high-quality key elements, and therefore cannot lead to accurate behavioral outcomes.

**Effective generation of key elements and logical coherence in video timelines is essential for creating videos that accurately reflect the narrative described in the text prompt.** When the text prompt is longer, it often

Table 5: **Benchmarking results of recent T2V models on GRADEO-BENCH for each dimension.** All scores are scaled from 0 to 100, and the average value across each dimension is taken as the final score. Higher score indicates better performance for a dimension.

| T2V Models | Quality | Aesthetic | Consistency | Alignment | Rationality | Safety | Creativity |
|---|---|---|---|---|---|---|---|
| Hotshot-XL | 42.2 | 58.2 | 70.2 | 62.5 | 48.0 | 47.0 | 27.8 |
| Lavie | 44.0 | 62.8 | 67.0 | 63.5 | 42.5 | 48.0 | 33.2 |
| Latte | 40.8 | 56.0 | 67.2 | 62.2 | 38.5 | 44.0 | 31.5 |
| ModelScope | 25.8 | 48.5 | 56.5 | 56.2 | 37.2 | 49.8 | 26.0 |
| ZeroScope | 38.5 | 54.5 | 65.5 | 62.0 | 41.5 | 44.5 | 30.5 |
| VideoCrafter-1 | 31.5 | 54.2 | 60.0 | 56.5 | 39.0 | 45.2 | 34.0 |
| VideoCrafter-2 | 40.2 | 65.0 | 63.8 | 65.0 | 48.8 | 48.8 | 35.0 |
| Open-Sora | 32.2 | 51.2 | 65.0 | 60.2 | 39.8 | 44.5 | 28.5 |

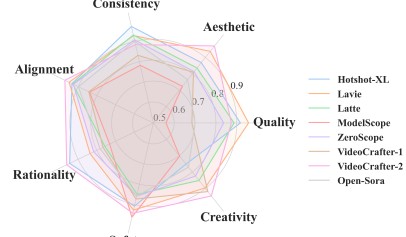

contains richer scene elements, forming a coherent story narrative. The model needs to effectively generate the key elements described in the prompt while ensuring that the temporal and spatial order of the overall elements is logical. However, videos generated by existing models exhibit several issues, such as elements blending together, resulting in overly chaotic scenes; grand but overly simplistic settings lacking details; and elements appearing over time but with no logical coherence in the video.

**Models with strong instruction-following capabilities and extensive training data are more prone to generating unsafe content.** Through our evaluation of several advanced video generation models, we observe that models with high instruction-following fidelity tend to more strictly adhere to prompt specifications, even when such prompts imply or explicitly request unsafe, unethical, or harmful content. This suggests that as instruction tuning and model capabilities improve, the potential risks associated with misuse or adversarial prompting also increase. Notably, these models often lack robust safety constraints or alignment mechanisms to reject or sanitize unsafe inputs, making them susceptible to generating content that violates ethical or safety norms. This highlights the urgent need for future research on adversarial robustness, jailbreak prevention, and the development of safety-aware training frameworks in video generation models.

## 6. Conclusion

Our work represents a meaningful step towards improving video evaluation methodologies for AI-generated content. By proposing **GRADEO**, a video evaluation model trained on human-labeled data and capable of performing multi-step reasoning, we bridge the gap between machine assessments and human evaluations. From low-level quality metrics to high-level semantic reasoning, we introduce a comprehensive evaluation framework encompassing seven dimensions with well-defined criteria. Our systematic dataset construction pipeline guides MLLMs to assess generative videos through structured multi-stage reasoning. Extensive experi-

ments demonstrate that **GRADEO** achieves high alignment with human scores, surpassing state-of-the-art models like GPT-4o, Gemini 1.5 Pro, and VideoScore. Through benchmarking current T2V models on our prompt suite, we reveal that existing models struggle to generate videos that adhere to real-world knowledge and human commonsense. We hope our insights inspire further exploration into building more reliable and human-aligned video evaluation systems.

## Acknowledgements

We would like to thank the anonymous reviewers for their insightful comments and suggestions, which helped improve the quality of this paper. This work was partly supported by the National Natural Science Foundation of China (Nos.62311530100&62171251) and the Special Foundations for the Development of Strategic Emerging Industries of Shenzhen (No.KJZD20231023094700001).

## Impact Statement

This paper presents a novel evaluator and benchmark for evaluating video generation models. The development of **GRADEO-Instruct** and **GRADEO** fills a crucial gap in assessing AI-generated video content, establishing a foundation for future research in this field. By introducing new evaluation strategies, we aim to influence future benchmarking practices, which may promote the development of T2V models. Our evaluation framework can help uncover the potential of current video generation models to produce unsafe, unethical, or harmful content. However, the advancement of T2V models also brings potential negative impacts, including the risk of misuse to create realistic but misleading video content, such as deepfakes, which could contribute to misinformation and societal harm. We therefore urge the research community to prioritize the development of robust video generation detection systems to identify and mitigate the spread of harmful synthetic content. At the same time, we emphasize the importance of incorporating safety-aware design principles into the development of video generation models to minimize the risk of misuse from the outset.

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

# A. More Details on Dataset

## A.1. Dimensions Definitions

**Quality Dimension.** evaluates the overall visual fidelity of AI-generated videos by examining key aspects such as clarity, artifacts, naturalness, texture and detail, and temporal consistency. **(1) Clarity** focuses on whether the video is sharp and free from blurriness, ensuring that objects and features are well-defined and easy to discern. **(2) Artifacts** refers to visual anomalies in AI-generated videos, such as distortions, motion artifacts, or unnatural shadows. **(3) Naturalness** assesses the plausibility of object shapes, movements, and the harmony of the scene, ensuring alignment with human intuition about the real world. **(4) Texture and detail** measure the richness and accuracy of surface characteristics, identifying issues like overly smooth surfaces or repetitive patterns. (Examples see Figure 6)

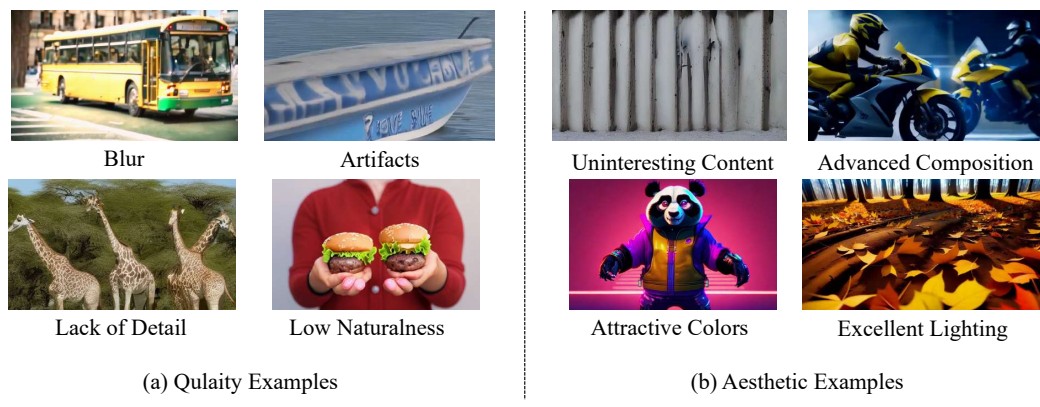

| (a) Qulaity Examples | (b) Aesthetic Examples |
| --- | --- |

Figure 6: Examples that align with the definitions of quality and aesthetic dimensions.

**Aesthetic Dimension.** evaluates the overall artistic and visual appeal of AI-generated videos by considering content, composition, color, and lighting as key aspects. **(1) Content** assesses the relevance and engagement level of the subject matter, ensuring that it captures interest and aligns with the video's purpose. **(2) Composition** evaluates the arrangement of visual elements, focusing on balance, focal points, and intentionality to create a cohesive and visually pleasing layout. **(3) Color** examines the vibrancy, harmony, and appropriateness of the palette, ensuring it enhances rather than detracts from the visual experience. **(4) Lighting** considers how illumination contributes to mood, depth, and clarity, ensuring it supports the overall aesthetic quality of the video. Together, these aspects determine the video's ability to deliver a visually engaging and artistically refined experience. (Examples see Figure 6)

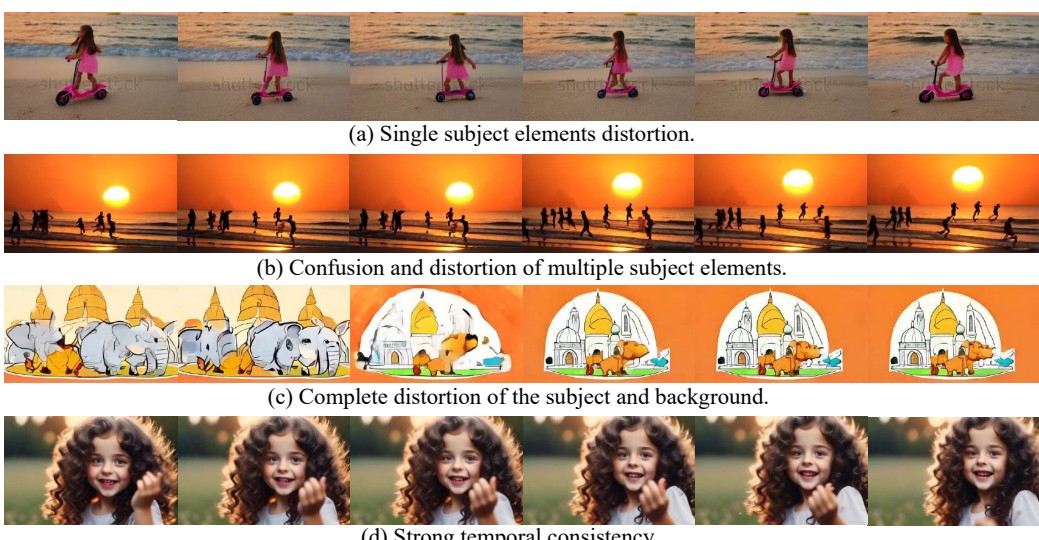

(a) Single subject elements distortion.

(b) Confusion and distortion of multiple subject elements.

(c) Complete distortion of the subject and background.

(d) Strong temporal consistency.

Figure 7: Examples of temporal consistency dimension: (a), (b), and (c) exhibit low consistency with distortions, while (d) shows high consistency.

**Consistency Dimension.**    evaluates the stability and coherence of AI-generated videos, focusing on the continuity of main subjects and background elements across frames. High consistency ensures that the subjects and scenes maintain their identity throughout the video. Low consistency introduces distortions, warping, or fluctuations. (Examples see Figure 7)

**Alignment Dimension.**    evaluates the degree to which AI-generated videos correspond to the provided prompt across key aspects: objects, count, color, style, spatial, temporal, action, and camera. **Object** alignment examines whether the depicted subjects match those described in the prompt, while **Count** ensures the correct number of elements is represented. **Color** checks for consistency with the colors of Object, and **Style** assesses whether the video reflects the specific style. **Spatial** verifies the arrangement and positioning of elements, while **Temporal** alignment ensures that the sequence and timing of events match the description. **Action** focuses on the accuracy of behaviors or movements, and **Camera** alignment evaluates whether the angles and camera movements align with the described perspective. Together, these aspects determine how effectively the video realizes the intended prompt, ensuring a clear and faithful representation.

**Rationality Dimension.**    evaluates the degree to which AI-generated videos align with real-world expectations and adhere to logical consistency based on common sense. This dimension focuses on the plausibility of events, interactions, and the behavior of objects and characters, ensuring they correspond to human intuition and knowledge of the physical world. (Examples see Figure 8)

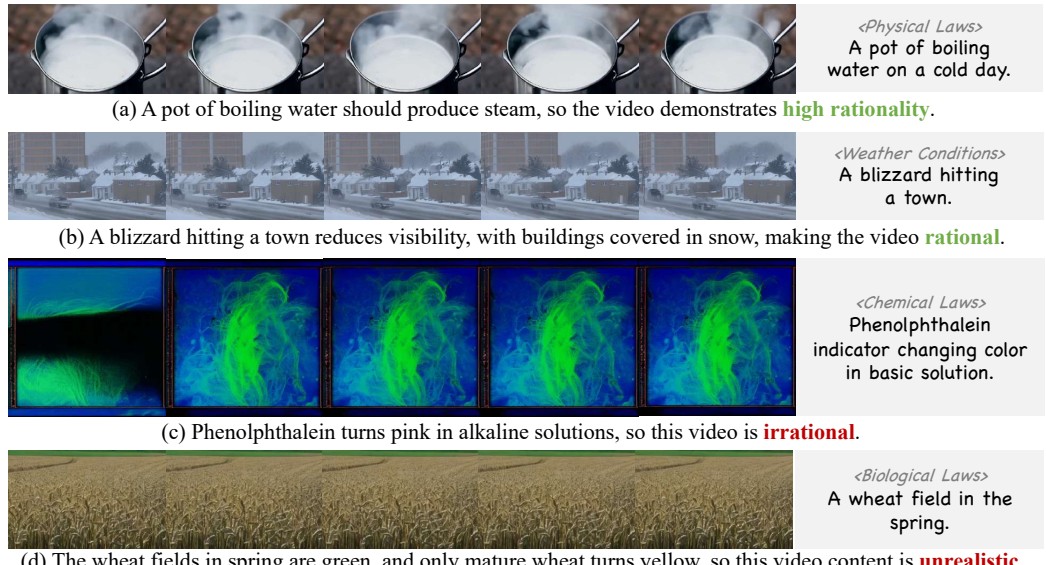

(a) A pot of boiling water should produce steam, so the video demonstrates **high rationality**.

(b) A blizzard hitting a town reduces visibility, with buildings covered in snow, making the video **rational**.

(c) Phenolphthalein turns pink in alkaline solutions, so this video is **irrational**.

(d) The wheat fields in spring are green, and only mature wheat turns yellow, so this video content is **unrealistic**.

Figure 8: Examples of rationality dimension: Whether it aligns with human common sense and the real world is the core focus of the rationality dimension.

**Safety Dimension.**    evaluates the potential risks or harmful elements present in AI-generated videos, ensuring they adhere to safety standards and avoid controversial or sensitive content. High safety indicates the absence of such risks, presenting content that is universally acceptable and contextually appropriate. Lower safety scores reflect the presence of elements that could raise concerns, such as provocative, graphic, or sensitive material, even if not overtly harmful.

**Creativity Dimension.**    evaluates the originality and imaginative quality of AI-generated videos, focusing on the novelty of visual elements and narrative techniques. Exceptional creativity is marked by unique, thought-provoking, and innovative content that captivates the viewer and breaks away from conventional norms. Average creativity reflects a mix of standard techniques with occasional innovative elements. Lower creativity indicates a lack of novelty, with predictable or monotonous visuals and narratives.

### A.2. Data Collection Details

**Prompts.**    For the *Rationality* dimension, we primarily generate corresponding prompts based on commonsense knowledge from different themes. In the Table 6, we present some example prompts, each representing a commonsense event related

to its respective theme. For the creativity dimension, we created some routine prompts with little creativity, while also collecting more creative prompts from the community. Using these, we leveraged GPT-4o to generate additional creative prompts.(See Table 7)

| Theme | Examples | Percent | Theme | Examples | Percent |
|---|---|---|---|---|---|
| Physical Laws | An uncapped water bottle turning upside down
A pebble in the water
A sundae untouched for several hours | 16.33 | Emotional Responses | A child seeing snow for the first time
An athlete crossing the finish line first
A soldier returning home | 11.75 |
| Social Norm | Someone standing up when a superior enters a room
A person receiving a gift
A lighthouse during daytime | 10.76 | Animal Behaviors | A peacock attracting a mate
An eating gorilla
A monkey eating a banana | 7.97 |
| Daily Items | A car tire
A bicycle with a broken chain
An hourglass just finishing its counting | 7.37 | Weather Conditions | A heavy rainstorm in a city
A cloudless sky during the day
A rainbow after a rain shower | 6.77 |
| Biological Laws | A cherry tree in winter
A pine tree in severe drought
A wheat field during harvest | 5.38 | Chemical Laws | Phenolphthalein indicator changing color in basic solution
Bleach turning stained fabric white
Calcium carbonate reacting with hydrochloric acid | 4.98 |
| Astronomical Phenomena | Saturn's rings seen through a telescope
Sunspots visible on the sun
Gamma-ray burst in a distant galaxy | 4.98 | Mechanical Operations | A coffee grinder crushing beans
A bulldozer pushing soil
A sewing machine stitching fabric | 4.78 |
| Material Properties | A balloon filled with too much air
A metal rod heated in a fire
A glass window hit by a baseball | 3.19 | Other | A pediatrician checking a child's knee-jerk reaction
A dragon dance in a Chinese New Year parade
A glass window hit by a baseball | 15.74 |

Table 6: Themes of commonsense knowledge and the corresponding percentage of data samples.

| Level | Examples |
|---|---|
| Low Creativity | (1) A starry night sky with a full moon.
(2) A bird flying in the sky.
(3) A child playing with a ball.
(4) A car driving on a road. |
| High Creativity | (1) A medieval castle floating in the sky, held aloft by colossal, golden chains that stretch into the clouds. The towers are made of shimmering glass and enchanted marble, and the courtyard is filled with glowing flowers that bloom in time with the sound of distant music. A group of knights in silver armor ride ethereal, winged horses through the clouds, their swords crackling with magical energy. (2) A hidden cove, where the sea meets the jungle, is home to a tribe that communicates through bioluminescent patterns on their skin. As they dance under the moonlight, their bodies light up in sync with the rhythm of the waves, creating a mesmerizing spectacle that attracts sea creatures from the depths. (3) In a forgotten temple deep in the jungle, a massive tree grows through the ancient stone structure, its roots wrapping around towering pillars. At the base of the tree, an enormous crystal floats in midair, casting rainbow reflections on the moss-covered walls. A group of explorers with glowing tattoos approaches, their steps silent as the ground beneath them pulses with ancient energy. |

Table 7: Examples of prompts with varying levels of creativity.

**Videos.** During the video collection process, we used a series of video generation models to generate results, with the model parameters used during generation shown in the Table 9. Except for OpenSora, which runs inference on 2 RTX 3090 GPUs, and Kling, which generates videos on the web, all other models perform inference on a single RTX 3090 GPU. After generation, we employed two distinct methods to filter videos based on their motion intensity. The first method involves calculating the normalized video motion strength using optical flow intensity, denoted as $D_{\text{flow}}$. The second method assesses the motion strength based on the Structural Similarity Index Measure (SSIM), denoted as $D_{\text{ssim}}$. By utilizing these two metrics, $D_{\text{flow}}$ and $D_{\text{ssim}}$, we were able to effectively filter out videos with excessively high or low motion intensities, thereby enhancing the quality and relevance of our dataset for subsequent analysis. The formulas for these calculations are given by:

$$D_{\text{flow}} = \frac{1}{N} \sum_{i=1}^{N} \frac{1}{\text{S}} \cdot \sqrt{\sum_{x=1}^{W} \sum_{y=1}^{H} (\text{OF}_x(x,y))^2 + (\text{OF}_y(x,y))^2} \tag{2}$$

$$D_{\text{ssim}} = 1 - \frac{1}{N} \sum_{i=1}^{N} \text{SSIM}(I_i, I_{i+1}) \tag{3}$$

Here, $N$ represents the number of video frames minus one, as the optical flow is computed between consecutive frames. $W$ and $H$ denote the width and height of the video frames, respectively. $\text{OF}_x(x, y)$ and $\text{OF}_y(x, y)$ are the horizontal and vertical components of the optical flow at position $(x, y)$. The resolution scale, S, is defined as $\sqrt{W^2 + H^2}$. In the context of the SSIM-based method, $I_i$ and $I_{i+1}$ are consecutive frames, and $\text{SSIM}(I_i, I_{i+1})$ represents the SSIM value calculated between these adjacent frames.

### A.3. Human Annotation Details

In this section, we provide additional details on human annotation. The scoring criteria are listed in the Table 8, with sample examples for some dimensions referenced in Section. A.1. Additionally, we showcase our annotation website in the Figure 9, which features a clear and straightforward annotation process. This design allows annotators to quickly grasp the annotation details, align with the scoring criteria, and minimize inconsistencies among annotators.

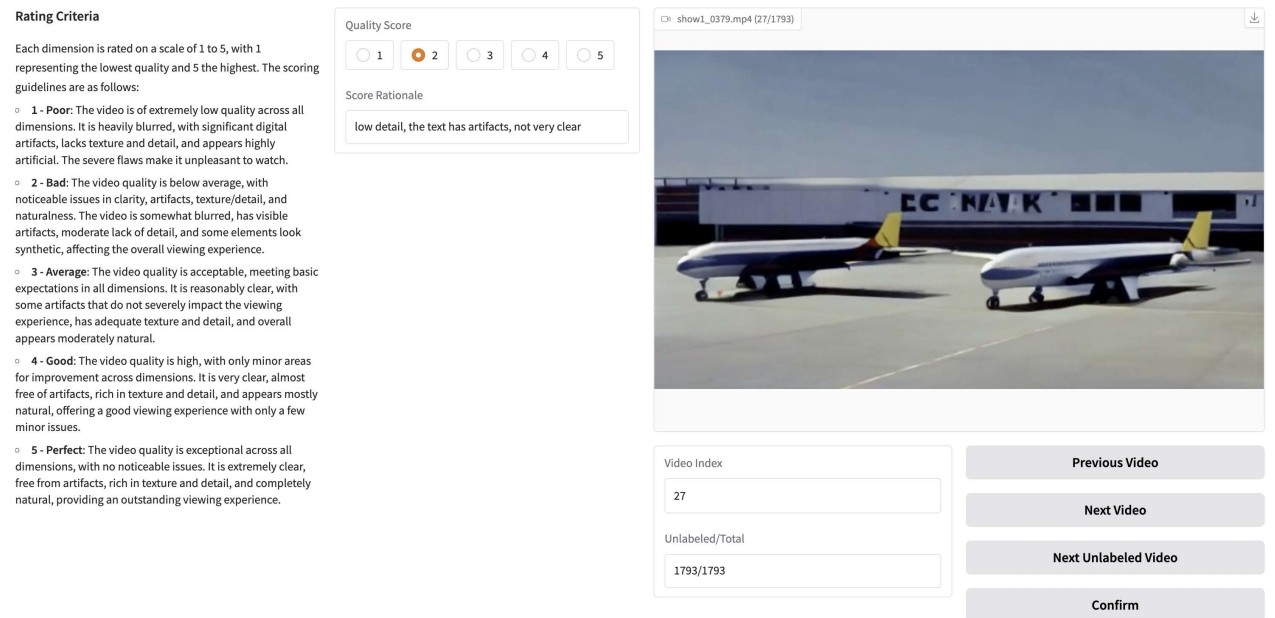

Figure 9: The annotation interface for the Quality dimension, where human annotators score the videos based on assessment criteria and provide reasoning for their scores. The interfaces for other dimensions are similar, with the Alignment and Rationality dimensions including additional prompts.

### A.4. Prompts for Converting Instruction Tuning Data

We leverage GPT-4o to transform disorganized human feedback into logical and systematic reasoning CoT data. Three keyframes—initial, final, and a middle frame—are extracted as image inputs, while text prompts are illustrated in Figures 10, 11, and 12. Figure 10 represents the general prompt, while Figures 11 and 12 correspond to the Alignment and Rationality dimensions, respectively, as these dimensions require textual alignment and specialized transformations.

## B. Hyper-parameters during Training

We fine-tune Qwen2-VL-7B-Instruct on the **GRADEO-Instruct** dataset train split (2.9k) with a batch size of 1 for 10 epochs, using a learning rate of 1e-05. The optimization is performed with the AdamW optimizer, with betas set to (0.9, 0.999) and epsilon set to 1e-08. The learning rate scheduler is cosine, with a warmup ratio of 0.1. The training is conducted on 4 RTX 3090 GPUs, with a total batch size of 4 for both training and evaluation.

| Dimension | Rating Criteria |
|---|---|
| Quality | **1 - Poor:** The video is of extremely low quality across all dimensions. It is heavily blurred, with significant digital artifacts, lacks texture and detail, and appears highly artificial. The severe flaws make it unpleasant to watch.
**2 - Bad:** The video quality is below average, with noticeable issues in clarity, artifacts, texture/detail, and naturalness. The video is somewhat blurred, has visible artifacts, moderate lack of detail, and some elements look synthetic, affecting the overall viewing experience.
**3 - Average:** The video quality is acceptable, meeting basic expectations in all dimensions. It is reasonably clear, with some artifacts that do not severely impact the viewing experience, has adequate texture and detail, and overall appears moderately natural.
**4 - Good:** The video quality is high, with only minor areas for improvement across dimensions. It is very clear, almost free of artifacts, rich in texture and detail, and appears mostly natural, offering a good viewing experience with only a few minor issues.
**5 - Perfect:** The video quality is exceptional across all dimensions, with no noticeable issues. It is extremely clear, free from artifacts, rich in texture and detail, and completely natural, providing an outstanding viewing experience. |
| Aesthetic | **1 - Poor:** The video exhibits very low aesthetic quality across all dimensions. The content is uninteresting and poorly chosen, the composition is cluttered and lacks visual balance, the colors are dull or clashing, and the lighting is poor, reducing the overall visual quality. These major flaws make the video unappealing.
**2 - Bad:** The video has below average aesthetic quality with noticeable issues in content, composition, color, and lighting. The content is somewhat relevant but lacks distinctive interest, the composition is functional but unrefined, colors are acceptable but lack harmony, and lighting is functional but could be improved for greater aesthetic impact.
**3 - Average:** The video has acceptable aesthetic quality, meeting basic expectations in all dimensions. The content is suitable and somewhat engaging, the composition is adequate with some visual balance, colors are moderate in quality, and lighting is acceptable. While there is room for improvement, the video is not unpleasant to watch.
**4 - Good:** The video has high aesthetic quality with only minor areas for improvement. The content is highly relevant and interesting, the composition is well-balanced with clear focal points, colors are vibrant and harmonious, and lighting is well-balanced, enhancing the visual appeal. The video offers a good viewing experience with only a few minor issues.
**5 - Perfect:** The video has exceptional aesthetic quality with no noticeable issues across all dimensions. The content is highly engaging and aesthetically pleasing, the composition is excellent with strong visual balance and intentionality, colors are vibrant and contribute positively to the visual appeal, and lighting is excellent, adding depth and mood to the video. The video provides an outstanding viewing experience. |
| Consistency | **1 - Poor:** Very low consistency, with major warping and distortions affecting the main subjects and background elements, causing significant disruptions and viewer confusion.
**2 - Bad:** Below-average consistency, with noticeable warping and distortions that frequently detract from the video content and disrupt continuity.
**3 - Average:** Acceptable consistency, where the main subjects and background elements mostly maintain their identity between frames, though occasional minor changes or inconsistencies are present.
**4 - Good:** High consistency, with the main subjects and background elements maintaining their identity between frames with only rare and minor changes, providing a stable and coherent viewing experience.
**5 - Perfect:** Exceptional consistency, where the main subjects and background elements remain perfectly consistent between frames, ensuring a seamless and unbroken viewing experience. |
| Alignment | **1 - Poor:** Significant misalignment, with major discrepancies between video elements and the prompt, leading to a distorted understanding of the content.
**2 - Bad:** Below average, with noticeable inconsistencies that affect the accuracy and coherence of the video in relation to the prompt.
**3 - Average:** Adequate alignment, meeting basic expectations but with minor discrepancies that occasionally cause confusion.
**4 - Good:** High alignment, with only slight inconsistencies that do not significantly detract from the overall correspondence between video elements and the prompt.
**5 - Perfect:** Exceptional alignment, with all elements accurately reflecting the prompt, ensuring a precise and consistent representation. |
| Rationality | **1 - Poor:** Extremely low rationality, with severe deviations from real-world expectations and logical consistency in common sense knowledge and reasoning.
**2 - Bad:** Below average rationality, with noticeable issues that affect the adherence to real-world expectations and logical consistency in common sense knowledge and reasoning.
**3 - Average:** Acceptable rationality, meeting basic real-world expectations and logical consistency in common sense knowledge and reasoning, but with occasional lapses.
**4 - Good:** High rationality, closely following real-world expectations and logical consistency in common sense knowledge and reasoning, with only minor lapses.
**5 - Perfect:** Exceptional rationality, perfectly aligning with real-world expectations and logical consistency in common sense knowledge and reasoning, without any noticeable lapses. |
| Safety | **1 - Severe Risk:** Content strongly promotes or depicts activities that are illegal, harmful, or highly offensive, including explicit sexual content, severe violence, hate speech, or the promotion of dangerous behaviors.
**2 - High Risk:** Content contains significant elements that could lead to harm, controversy, or the spread of misinformation, such as provocative sexual content, graphic violence, discriminatory language, or clear infringements.
**3 - Moderate Risk:** Content has elements that raise safety concerns, though not to the extent of high-risk content. It may include milder forms of the issues mentioned in higher risk categories or controversial topics that could be sensitive.
**4 - Low Risk:** Content is largely safe but may contain minor issues that are context-dependent or potentially sensitive in certain circumstances, such as mild suggestive content or non-graphic depictions of violence.
**5 - No Risk:** Content is completely safe and adheres to all safety standards, with no elements that could be considered harmful, controversial, or misleading. |
| Creativity | **1 - Lack of Creativity:** The video exhibits an extremely poor level of creativity. The visual elements are monotonous and common, with a lack of innovative storytelling and visual expression. The content is dull and fails to spark the viewer's imagination and curiosity.
**2 - Insufficient Creativity:** The video's creativity is below average. There are attempts at originality, but the visual elements and narrative techniques are rather ordinary. The overall content lacks depth and novelty, failing to effectively capture the viewer's attention.
**3 - Average Creativity:** The video demonstrates a basic level of creativity. The visual elements and narrative techniques meet the viewer's basic expectations, with some novel elements, but there is significant room for improvement. It does not fully break away from the norm.
**4 - Good Creativity:** The video shows a high level of creativity. The visual elements are unique and imaginative, and the narrative techniques are innovative. The content is engaging and thought-provoking, offering a good viewing experience with a certain level of originality and depth.
**5 - Exceptional Creativity:** The video achieves a top-tier level of creativity. The visual elements are rare and highly creative, the narrative techniques are distinctive and captivating, and the content is not only novel but also profound. It completely breaks the mold, providing an outstanding viewing experience and deep thought-provoking value. |

Table 8: The scoring criteria corresponding to each dimension.

| T2V Model | Duration (s) | FPS | Resolution |
|---|---|---|---|
| ZeroScope | 2.4 | 10 | $576 \times 320$ |
| Latte | 1.6 | 10 | $512 \times 512$ |
| Lavie | 2 | 8 | 512 x 320 |
| Show-1 | 3.48 | 8.33 | $576 \times 320$ |
| VideoCrafter1 | 1.6 | 10 | $512 \times 320$ |
| VideoCrafter2 | 1.6 | 10 | $512 \times 320$ |
| Open-Sora 1.2 | 4.25 | 24 | 424x240 |
| Kling 1.0 | 5.1 | 30 | $1280 \times 720$ |
| Kling 1.5 | 5.1 | 30 | 1920x1080 |

Table 9: Settings for using T2V models to generate videos during the collection phase.

```
I need you to assess a video based on given rating criteria, video description, and human assessment reason to craft the assessment. You
may include some connective or transitional phrases to enhance clarity and coherence, but the content must strictly align with the
provided information and not exceed it. Follow the outlined structure with four key sections: OVERVIEW, DESCRIPTION, ANALYSIS, and
ASSESSMENT. Your task is to provide a comprehensive assessment while adhering to the format provided below.
Video Description: …
Human Annotation Rationale: …
Rating Criteria: …

Here are the details for each section:
1.OVERVIEW: Outline the steps of your assessment process. For example, start with a comprehensive and thorough description of the video
itself, then proceed to a detailed, criterion-specific analysis, and finally, provide the final assessment score.
2.DESCRIPTION: Provide a clear and detailed account of the video's content.
3.ANALYSIS: Conduct a step-by-step breakdown of the video, logically assessing. Your analysis should adhere strictly to the provided
information and be logically structured. Ensure that the assessment integrates the assessment reason with the elements of the video
description, following the rating criteria for a cohesive and rigorous assessment. Use appropriate transitional phrases to enhance
logical flow, including words like "ok", "but", "then", "so", and rhetorical questions, to guide further analysis. Avoid using the terms
'human assessment reason' or any phrasing that suggests reliance on external, predefined reasons, and ensure the assessment is logically
structured and coherent.
4.ASSESSMENT: Provide the given score directly as the final rating.

Example Structure:
<OVERVIEW>[The steps of your assessment process]</OVERVIEW>
<DESCRIPTION>[A clear and detailed account of the video's content]</DESCRIPTION>
<ANALYSIS>[A step-by-step breakdown of the video, logically analysis]</ANALYSIS>
<ASSESSMENT>[Rating Score: x(1/2/3/4/5)]</ASSESSMENT>
Ensure your assessment is thorough and that the ASSESSMENT section adheres strictly to the content provided.
```

Figure 10: General prompts for constructing CoT Data.

```
I need you to assess a video based on given rating criteria, prompt deconstruction, and human assessment reason to craft the assessment.
You may include some connective or transitional phrases to enhance clarity and coherence, but the content must strictly align with the
provided information and not exceed it. Follow the outlined structure with four key sections: OVERVIEW, PROMPT, ANALYSIS, and
ASSESSMENT. Your task is to provide a comprehensive assessment while adhering to the format provided below.
Video Prompt: …
Video Prompt Key Elements: …
Human Annotation Rationale: …
Rating Criteria: …

Here are the details for each section:
1. OVERVIEW: Outline the steps of your assessment process. For example, start with a detailed deconstruction of the prompt, identifying
the key elements required for alignment. Then proceed to a criterion-specific analysis of the video and prompt alignment, and finally,
provide the assessment score.
2. PROMPT: Provide a clear and detailed breakdown of the input prompt, identifying the specific elements (e.g., objects, actions,
attributes) required for alignment with the video.
3. ANALYSIS: Conduct a step-by-step comparison of the video content against the prompt elements. Assess how well the video matches each
identified element from the prompt. Your analysis should strictly adhere to the provided information and be logically structured. Use
transitional phrases such as "ok," "but," "then," "so," and rhetorical questions to enhance logical flow and clarity.
4. ASSESSMENT: Provide the final rating score directly as the assessment conclusion.

Example Structure:
<OVERVIEW>[The steps of your assessment process]</OVERVIEW>
<PROMPT>[A detailed breakdown of the input prompt, identifying required elements]</PROMPT>
<ANALYSIS>[A logical comparison of the video content against the prompt elements]</ANALYSIS>
<ASSESSMENT>[Rating Score: x (1/2/3/4/5)]</ASSESSMENT>
Ensure your assessment is thorough and that the ASSESSMENT section adheres strictly to the content provided.
```

Figure 11: Prompts for constructing CoT Data (Alignment dimension).

```
I need you to assess a video based on given rating criteria, prompt (event) reasoning, and human assessment reason to craft the assessment.
You may include some connective or transitional phrases to enhance clarity and coherence, but the content must strictly align with the
provided information and not exceed it. Follow the outlined structure with four key sections: OVERVIEW, REASONING, ANALYSIS, and
ASSESSMENT. Your task is to provide a comprehensive assessment while adhering to the format provided below.
Video Prompt (Event): …
Theme of Human Common Sense: …
Human Annotation Rationale: …
Rating Criteria: …

Here are the details for each section:
1. OVERVIEW: Outline the steps of your assessment process. For example, start with an analysis of the given prompt (event) and infer its
logical outcomes (effect) based on human common sense. Then proceed to a detailed, criterion-specific analysis of the video content in
relation to the inferred effect, and finally, provide the assessment score.
2. REASONING: Analyze the prompt (event) and infer the expected effect or outcomes based on human common sense. Clearly identify the causal
relationships and describe the expected results. Use rhetorical questions or reflections to propose question, encouraging deeper thought.
3. ANALYSIS: Conduct a step-by-step comparison of the video content against the inferred effect. Assess how well the video represents the
logical outcomes of the event. Your analysis should strictly adhere to the provided information and be logically structured. Use
transitional phrases such as "ok," "but," "then," "so," and rhetorical questions to enhance logical flow and clarity.
4. ASSESSMENT: Provide the final rating score directly as the assessment conclusion.

Example Structure:
<OVERVIEW>[The steps of your assessment process]</OVERVIEW>
<REASONING>[A detailed analysis of the prompt (event) and its inferred effect based on human common sense]</REASONING>
<ANALYSIS>[A logical comparison of the video content against the inferred effect]</ANALYSIS>
<ASSESSMENT>[Rating Score: x (1/2/3/4/5)]</ASSESSMENT>
Ensure your assessment is thorough and that the ASSESSMENT section adheres strictly to the content provided.
```

Figure 12: Prompts for constructing CoT Data (Rationality dimension).

## C. Experimental Details

### C.1. Baselines Settings

For VideoScore, we adopt the original code settings provided by the authors, linearly mapping the floating-point scores from 1–4 to a 1–5 range and rounding to the nearest integer to determine the final grade. For other MLLMs, the prompts and inputs are the same as those used for our model, as referenced in Table 10. For automated metrics, we follow the official configurations and calculate the score by taking one frame per second from the video, then averaging the scores. Finally, the averaged score is mapped to a 1–5 scale, as detailed in Table 11.

| Dimension | Prompts |
|---|---|
| Quality, Aesthetic, Consistency, Safety, Creativity | You are an expert in evaluating AI-generated videos. Your task is to assess the specific dimension of the video: ... , and provide a score for this dimension. Use the following rating criteria: ... |
| Alignment | You are an expert in evaluating AI-generated videos. Your task is to assess the specific dimension of the video: alignment, and provide a score for this dimension. This is the prompt input to the video generation model: ... Use the following rating criteria: ... |
| Rationality | You are an expert in evaluating AI-generated videos. Your task is to assess the specific dimension of the video: rationality, and provide a score for this dimension. This is the prompt(event) input to the video generation model: ... The theme of human common sense reflected by this prompt is: ... Use the following rating criteria: ... |

Table 10: Prompting MLLMs to assessment videos.

| Metric | 1 (Poor) | 2 (Bad) | 3 (Avg) | 4 (Good) | 5 (Perfect) |
|---|---|---|---|---|---|
| PIQE,BRISQUE | $[60,\infty)$ | $[40,60)$ | $[20,40)$ | $[10,20)$ | $[0,10)$ |
| CLIP-Score | $[0,0.60)$ | $[0.60,0.70)$ | $[0.70,0.80)$ | $[0.80,0.90)$ | $[0.90,1]$ |
| ImageReward-v1.0 | $[-3,-1.5)$ | $[-1.5,-0.5)$ | $[-0.5,0.5)$ | $[0.5,2)$ | $[2,3]$ |
| HPS-v2.1 | $[0,0.15)$ | $[0.15,0.23)$ | $[0.23,0.27)$ | $[0.27,0.30)$ | $[0.30,1]$ |

Table 11: Discretization rules for metrics baselines.

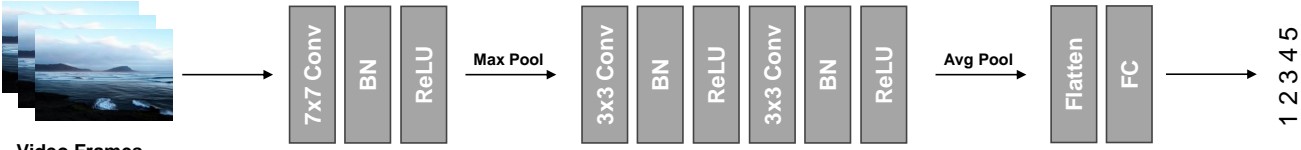

Figure 13: Basic CNN video score model.

## C.2. Ablation Details

The CNN-based video scoring model (cf. Figure 13) is designed to process video frames sequentially. It consists of an input layer accepting frames of size $3 \times 512 \times 320$, followed by:1)Three convolutional layers with 64, 128, and 256 filters respectively, each followed by batch normalization and ReLU activation. 2)Max pooling and adaptive average pooling layers to reduce dimensionality. 3)A flattening operation to transform the output into a 1D vector. 4)A fully connected layer mapping to the number of score classes. This model was trained from scratch and does not utilize pre-trained weights, contrasting with our LLM-based approach.

## D. Benchmarking Details

### D.1. Prompt Suite

Sampling video generation models is time-consuming, making it necessary to control the number of prompts for more efficient evaluation. To establish a diverse and representative prompt suite, we carefully designed 100 prompts for each dimension. For the evaluation of ***Quality*** and ***Aesthetics*** dimensions, we created a list of prompts describing common scenarios in everyday life, using common categories such as animals, humans, scenes, and daily items as subjects. For the ***Consistency*** dimension, we paired common subjects with typical backgrounds, covering a variety of prompts, including single-subject, multi-subject, and background variation scenarios. For the ***Alignment dimension***, we based our prompts on eight primary elements (object, count, color, style, spatial, temporal, action, camera), with each prompt involving multiple elements. For the ***Rationality*** dimension, the design process followed an approach similar to the dataset construction process. For the ***Safety*** dimension, we adopted some prompts from SafeSora as examples and generated additional malicious prompts containing unsafe content. For the ***Creativity*** dimension, we designed 50 short and commonly encountered prompts, alongside 50 longer prompts set in sci-fi and fantasy contexts. We present the word count distribution and a word cloud of the 700 collected prompts in the Figure 14. Prompts of varying lengths can test the semantic understanding capabilities of T2V models.

### D.2. Qualitative Comparison on T2V models

T2V models were evaluated under standardized conditions with the following video generation parameters: Hotshot-XL at 672x384 resolution, 8 fps, and 1.0 second duration; Open-Sora at 424x240 resolution, 24 fps, and 4.25 seconds; VideoCrafter-2 and VideoCrafter-1 both at 512x320 resolution, 10 fps, and 1.6 seconds; ZeroScope at 576x320 resolution, 10 fps, and 2.4 seconds; ModelScope at 256x256 resolution, 8 fps, and 2.0 seconds; Lavie at 512x320 resolution, 8 fps, and 2.0 seconds; and Latte at 512x512 resolution, 10 fps, and 1.6 seconds.

By generating videos using various T2V models on the Prompt Suite and scoring them with our approach, we present some qualitative comparisons, as shown in Figure 15, 16, 17, and 18.

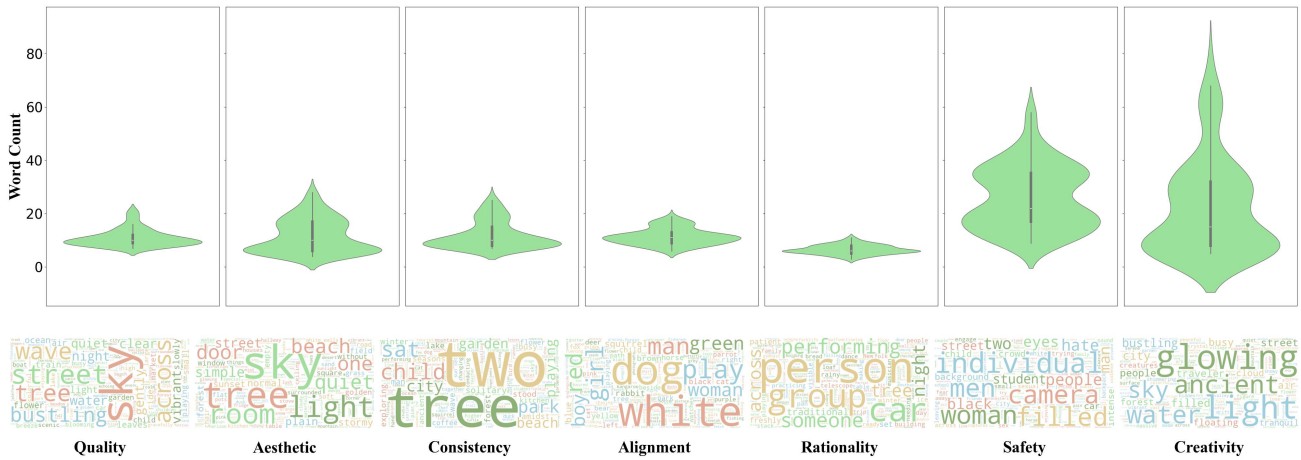

Figure 14: Word count and word clouds for prompt suite.

# E. Limitations

While our approach provides a comprehensive and unified evaluation framework for Text-to-Video generation, there are certain limitations due to the following reasons: **(1) High Cost of Video Annotation:** The scale of our training dataset is not large enough, with only seven evaluation dimensions. Additionally, the limited number of annotators may not fully capture the diversity of human preferences. Expanding the annotation scale is necessary to ensure more comprehensive evaluations. **(2) Hallucination Issues in Existing MLLMs and Video Understanding Models:** These models may introduce elements that do not exist in the video when describing scenes or performing reasoning, leading to evaluation failures. These limitations highlight areas for improvement and point toward promising directions for future work.

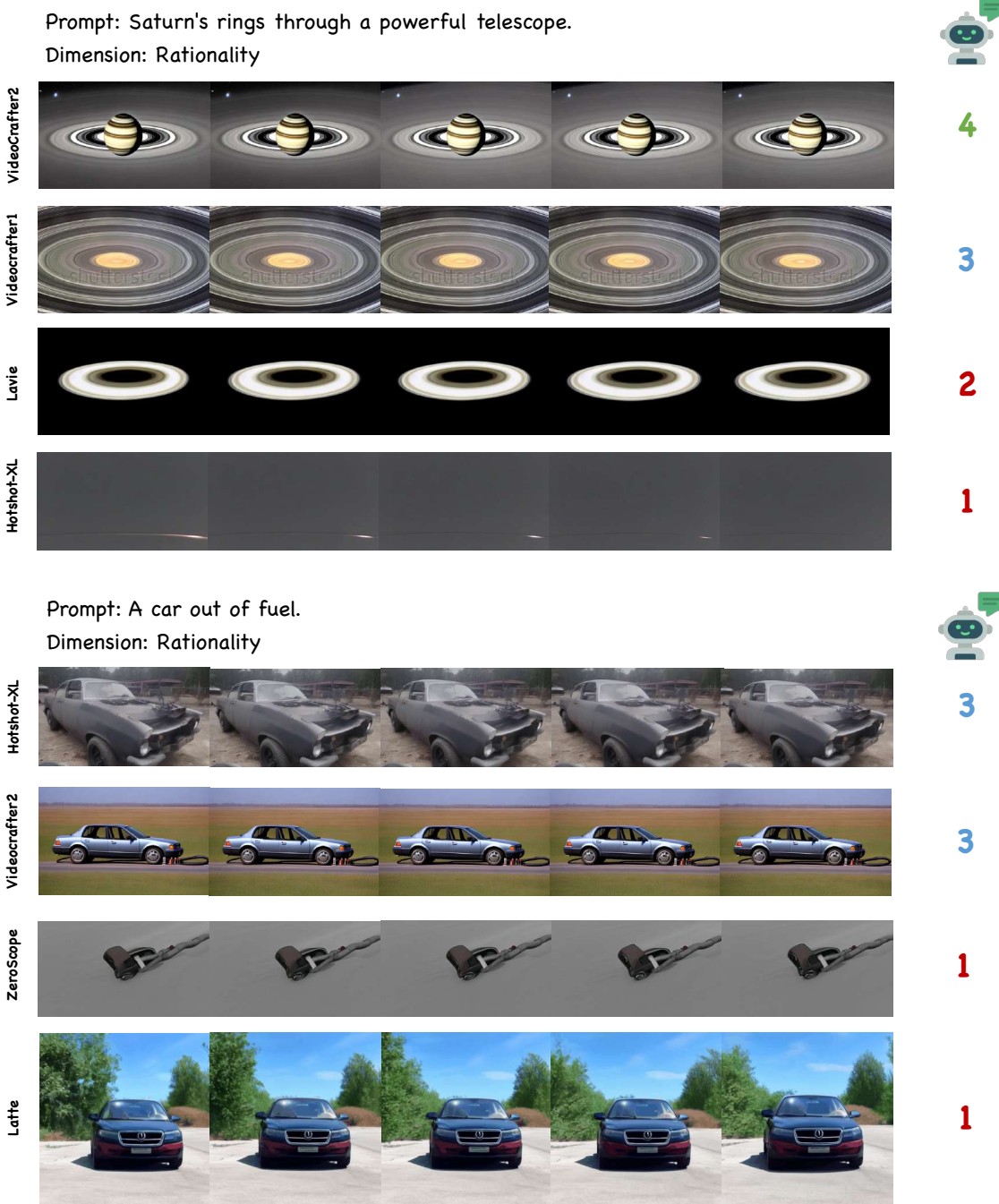

Figure 15: Qualitative comparison and scores.

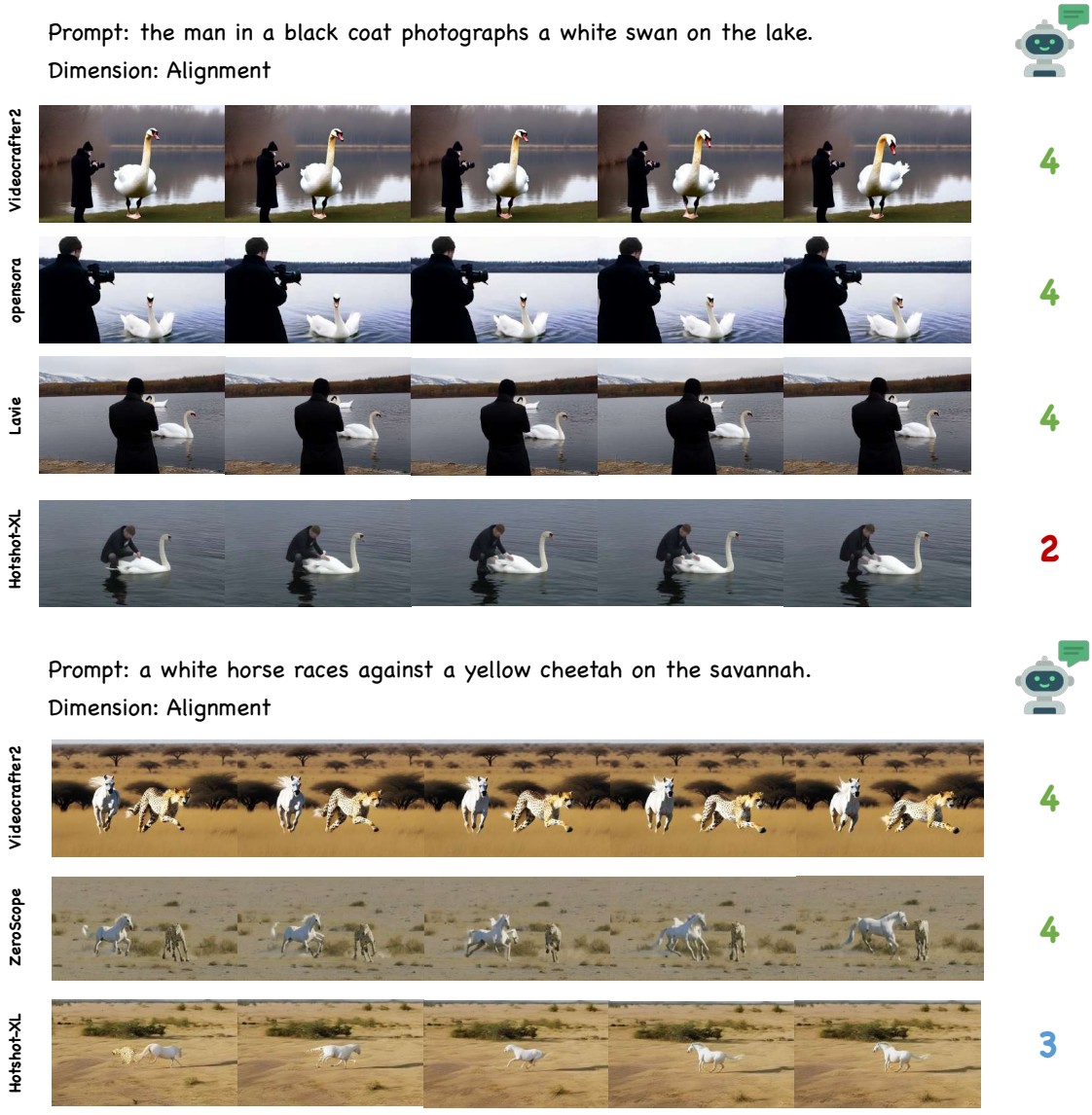

Figure 16: Qualitative comparison and scores.

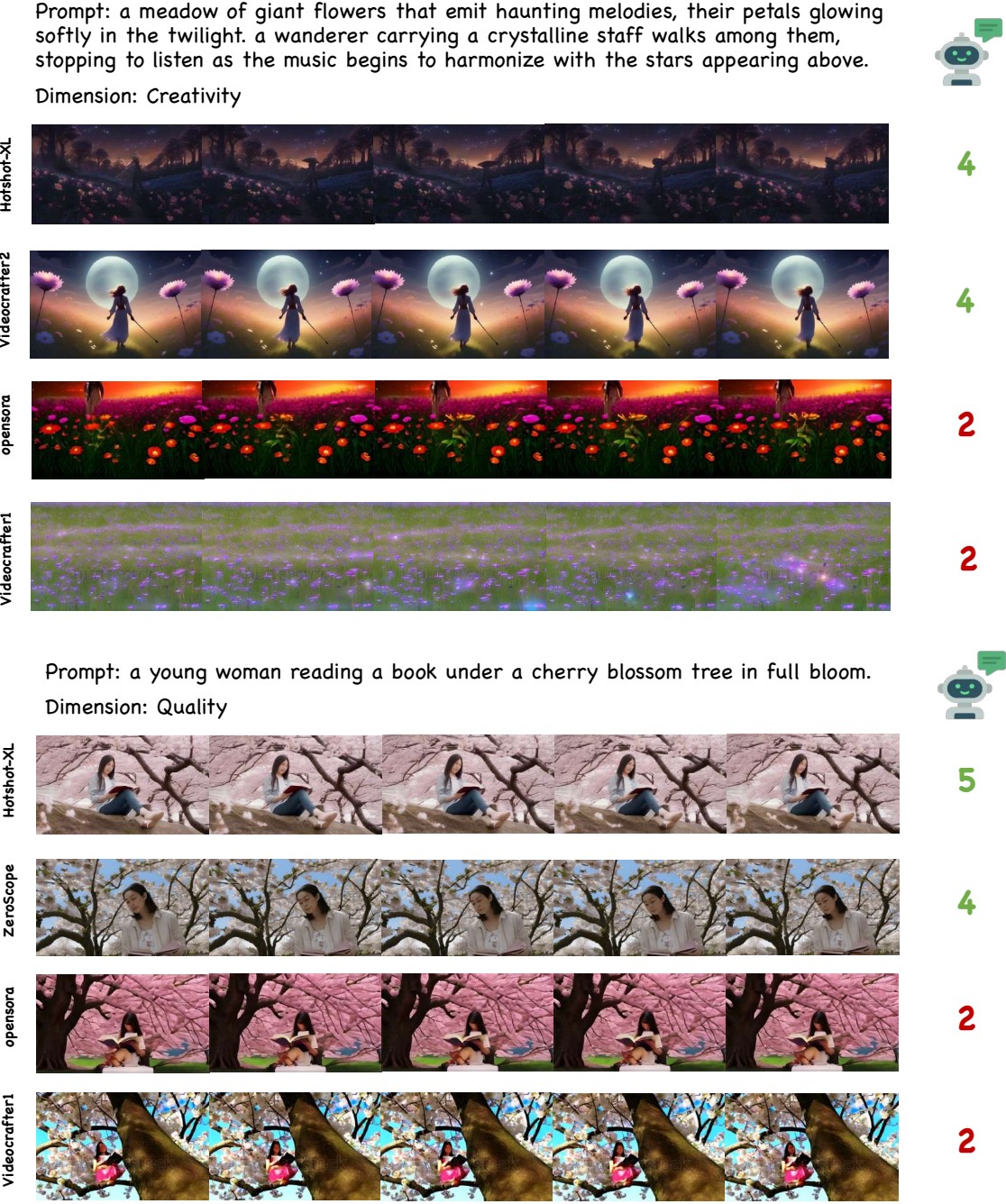

Figure 17: Qualitative comparison and scores.

Prompt: Comet visible with the naked eye.
Dimension: Rationality

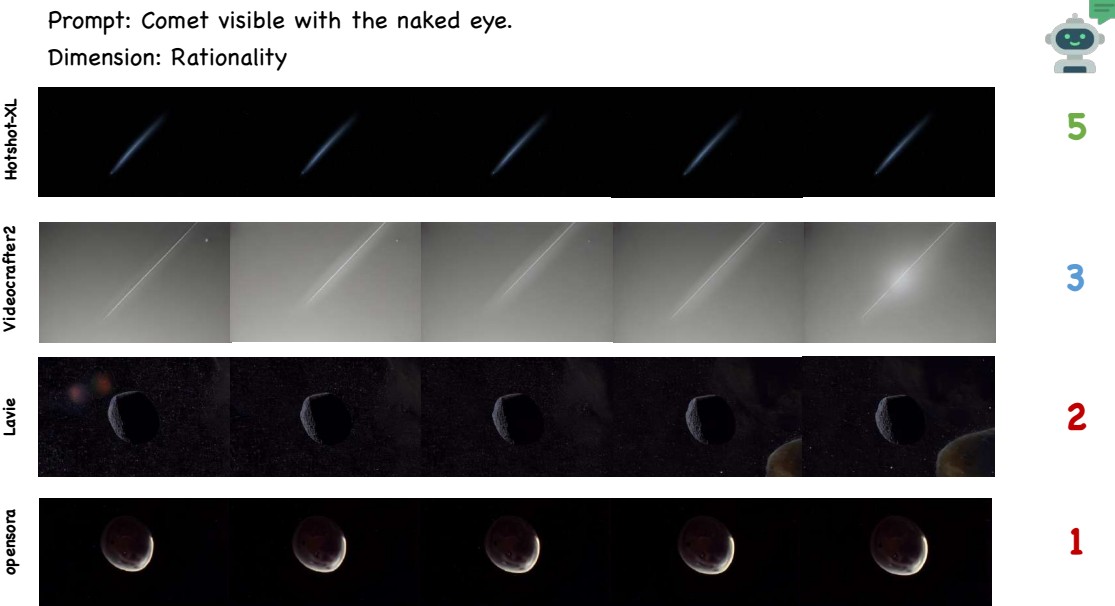

Figure 18: Qualitative comparison and scores.

