# OpenReview forum: "GRADEO: Towards Human-Like Evaluation for Text-to-Video Generation via Multi-Step Reasoning"
_ICML.cc/2025/Conference — ICML 2025 poster_

### Official Review · Reviewer_RczA · 2025-03-02

**Overall Recommendation:** 3

**Summary:**

The paper introduces GRADEO, a model for evaluating text-to-video generation using multi-step reasoning. The authors propose GRADEO-Instruct dataset, which contains 3.3k videos and 16k human annotations, to train GRADEO to mimic human evaluation. The evaluation metrics include multiple dimensions, including quality, aesthetic, consistency, alignment, rationality, safety, and creativity. The paper evaluate various T2V models by the proposed GRADEO and shows that GRADEO aligns better with human evaluations than existing methods.

---

**[Update after Rebuttal] Final Comment by Reviewer RczA**

I thank the authors for providing a detailed response. Overall, this paper explores an interesting idea by incorporating multi-step reasoning into T2V evaluation.

However, the rebuttal does not fully address my concern about the limited scale of benchmark dataset, which was also pointed out by Reviewer hvju. Specifically, the table included in the rebuttal clearly shows that the model performance improves with increasing more training data, indicating the current dataset size is insufficient. While the authors explained that it is limited by resources, I believe there is potential to explore automated pipelines to scale up data collection more efficiently.

As such, I will maintain my original rating as weak accept.

**Claims And Evidence:**

- As one of the motivation of GRADEO, the authors claim that the current LLM-based evaluation methods only provide scores without rationale, which is a reasonable claim and provides an inspirefeul insight for involving the idea of multi-step reasoning in T2V evaluation.

- The authors claim that current T2V models struggle with high-level semantic understanding. Such weaknesses is confirmed by the proposed benchmark, mentioning that most of the state-of-the-art models perform poorly in rationality and creativity scores.

**Essential References Not Discussed:**

This paper has well cited most of the related works.

**Experimental Designs Or Analyses:**

- The experiments are comprehensive. To evaluate the proposed GRADEO evaluator model, the authors provide the correlation analysis with human scores and compare with a wide variety of the models. Such experiments are also conducted on multiple datasets.

- In the while, the authors also benchmarks the current T2V models using the proposed GRADEO evaluator model and provide some insights (e.g., most models perform poorly in creativity and rationality) based on the results.

**Methods And Evaluation Criteria:**

- To fine-tune a LLM for video evaluation, the authors collected a GRADEO-Instruct dataset with 3.3k videos and 16k human annotations. However, I doubt that such scale of the dataset is large enough to train a good evaluator. It would be interesting to see an experiment ablating different scales of training dataset (e.g., including 1k / 2k / 3k videos for training) and check how the dataset scale affect the model performance.

- GRADEO-Instruct dataset only includes AI-generated videos, limiting its perspective and knowledge on synthetic videos. Most of the LLM-based methods, such as VideoScore, include both real and fake videos for training, which is a more reasonable setting and could teach the model distinguish the videos from these two distributions.

- The evaluation criteria is comprehensive and well-structured. It includes seven evaluation dimensions, covering both low-level perception (e.g., quality, aesthetic) and high-level semantic reasoning (e.g., rationality, creativity).

[1] "VIDEOSCORE: Building Automatic Metrics to Simulate Fine-grained HumanFeedback for Video Generation"

**Other Comments Or Suggestions:**

- Line 305, with those of "GRADE" -> there is a missing "O"

**Other Strengths And Weaknesses:**

- The overall presentation is good. The paper is well written and the figures are properly plotted which could help readers quickly understand the proposed method.

**Questions For Authors:**

- Have the authors considered adopting automatic pipeline to expand the dataset scale with a smaller cost?

- What is the model architecture of GRADEO? Since GRADEO model is one of the main contribution of this paper, it would be better to elaborate how the model backbone is designed. For example, how is the video encoded and inputted to the proposed vision-language model? Different model designs could also significantly affect the model performance.

**Relation To Broader Scientific Literature:**

The paper introduces a novel and well-motivated T2V evaluation model by loosening the constraint that the current evaluation model usually provides the scores only without the rationale behind. Through annotating the videos with the evaluation of multi-step reasoning, the model performance could further be improved from the chain-of-thoughts reasoning.

**Theoretical Claims:**

There is no theoretical claims in this paper.

---

> ### Author Rebuttal · Authors · 2025-03-28
>
> We sincerely thank you for your comprehensive comments and constructive advice. We are very excited to see that the reviewer finds our work (1)"provides an **inspireful insight**", (2) "evaluation criteria is **comprehensive and well-structured**", (3)"**comprehensive experiments**", (4)and "paper is well written". Your suggestions and questions are valuable, and we will explain your concern as follows.
>
> **[Q1] Concerns about the size of the dataset.**
>
> **[A1]** We conducted the following experiments and analyses to allay your concerns as much as possible. We trained on different amounts of training data and tested on the test set. The similarity and consistency between the dimensions and the human evaluation are in the table below. We found that for the model trained on 1k, 2k data, the model is able to learn the entire format of the assessment, the output is formatted correctly. There is a performance improvement as data increases.
>
> |  | Quality | Aesthetic |  Consistency | Alignment | Rationality |  Safety | Creativity |
> | --- | --- | --- | --- | --- | --- | --- | --- |
> | 1k | 0.589/0.533/0.526 | 0.697/0.666/0.404 | 0.502/0.238/0.500 | 0.549/0.511/0.821 | 0.552/0.513/0.633 | 0.727/0.765/0.360 | 0.675/0.654/0.792 |
> | 2k | 0.626/0.600/0.474 | 0.667/0.619/0.333 | 0.565/0.414/0.477 | 0.560/0.497/0.786 | 0.551/0.510/0.571 | 0.712/0.756/0.380 | 0.771/0.756/0.792 |
> | all | 0.743/0.715/0.404 | 0.717/0.719/0.351 | 0.634/0.641/0.341 | 0.601/0.418/0.439 | 0.606/0.515/0.560 | 0.747/0.762/0.360 | 0.797/0.759/0.542 |
>
> In fact, we believe that orders of magnitude larger (10k or even 100k) video and human assessment reason data and a better training approach (as opposed to cost-saving LoRA fine-tuning) may be able to result in a more robust t2v evaluation model. Our work has been constrained by GPU computational resources and funding limitations, preventing further expansion. In addition to this, **some of the previous AIGC quality assessment datasets are also not very large**; the T2VQA-DB [1] dataset contains 10,000 videos, the MQT [2] dataset has only 10,005 videos, the FETV [3] dataset has only 2,476 videos, and the LGVQ [4] dataset has only 2,808 videos and the above datasets only contain human subjective ratings, which are not available for the training of interpretable models.
>
> **[Q2] Typo error.**
>
> **[A2]** Thank you for your careful reading. We will correct this issue and carefully check for any other typo errors in the revised version of the paper.
>
> **[Q3] Automatic pipeline to expand the dataset scale.**
>
> **[A3]** Your comments are very insightful. We used an automated approach based on GPT-4o in generating the instruction tuning dataset from human assessment reasons, reducing the cost of organizing human assessment reasons into assessment CoT data. The main limitation to our dataset construction is human annotations. Instruction tuning for MLLMs requires human-assessed instruction datasets, and collecting human assessments for each video is time-consuming and costly. In order to minimize the impact of annotators' own biases on the evaluation dataset, we exclude videos from the dataset if there is a significant disparity between annotators' ratings. Overall, human evaluation are the **gold standard** and proof of the credibility of the assessment model. Collecting human assessment reason is a step that we believe cannot be replaced by an automated approach, which allows the **high dataset construction cost to limit the size of the dataset**. We also hope for more efficient methods to scale human annotation in the future.
>
> **[Q4] Model architecture.**
>
> **[A4]** We adopt Qwen2-VL-7B[5] as our base model. Qwen2-VL-7B adopts the tandem structure of ViT and Qwen2, and mainly realizes two major innovative architectures: (1) Fully supports native dynamic resolution, which can handle image inputs of arbitrary resolution, and different sizes of images are converted into dynamic numbers of tokens, with a minimum of only 4. (2) Proposes M-ROPE to decompose rotational position embedding decomposition into three parts: time, height, and width, so that the large language model can integrate the position information of one-dimensional text, two-dimensional image and three-dimensional video.
>
> [1] Subjective-aligned dataset and metric for text-to-video quality assessment. ACM-MM 2024.
>
> [2] Measuring the quality of text-to-video model outputs: Metrics and dataset.
>
> [3] FETV: A Benchmark for Fine-Grained Evaluation of Open-Domain Text-to-Video Generation.NeurIPS 2023
>
> [4] Benchmarking Multi-dimensional AIGC Video Quality Assessment: A Dataset and Unified Model.
>
> [5] Qwen2-VL: Enhancing Vision-Language Model's Perception of the World at Any Resolution.
>
> ---
>
> Finally, we deeply appreciate the thoughtful questions and suggestions, which have greatly contributed to improving our work and will be incorporated into our revised manuscript. **We sincerely hope it sufficiently address your concerns and earn your recognition.**

---

> > ### Comment · Reviewer_RczA · 2025-04-07
> >
> > I thank the authors for providing a detailed response. Overall, this paper explores an interesting idea by incorporating multi-step reasoning into T2V evaluation.
> >
> > However, the rebuttal does not fully address my concern about the limited scale of benchmark dataset, which was also pointed out by Reviewer hvju. Specifically, the table included in the rebuttal clearly shows that the model performance improves with increasing more training data, indicating the current dataset size is insufficient. While the authors explained that it is limited by resources, I believe there is potential to explore automated pipelines to scale up data collection more efficiently.
> >
> > As such, I will maintain my original rating as weak accept.

---

> > > ### Author Response · Authors · 2025-04-07
> > >
> > > Thanks for your reply. We would like to further clarify that our training dataset is sufficient to fully stimulate the model's video evaluation ability. To address your concern, we respond from the following two perspectives:
> > >
> > > - (1) The table presented in the previous rebuttal A1 was too coarse-grained to effectively demonstrate how performance improves as the data volume increases, which may have caused some confusion. To address this, we provide comparisons of SROCC and PLCC across a broader range of training data amounts, presented through a curve plot (Figure 1: SROCC (↑) and PLCC (↑) with respect to varying training dataset sizes. Both metrics improve with larger datasets, though the performance gains gradually plateau as the dataset size increases. Anonymous link: https://imgur.com/a03N3iG) along with a Table 1, showing the performance growth clearly. It can be noticed that around 3K data, the model performance improvement with the increase of data has become relatively small, so we chose 3.3K data to be a good balance between the cost of data collection and the performance of the model. Therefore, we believe that our 3.3K data are sufficient to stimulate the model's video evaluation ability, especially given that the marginal effect of adding more data becomes increasingly negligible.
> > >
> > > - (2) We would like to emphasize that **our work introduces one of the first human preference-based video evaluation datasets, aiming to teach the model human-like video evaluation capabilities, especially in high-level semantic understanding and reasoning**. Our work establishes a foundation and benchmark in this area, upon which future research can build to further advance this important and promising direction. Notably, after aligning with human preferences using our data, our model's video evaluation capabilities significantly surpass SOTA MLLMs, such as GPT-4o and Gemini-1.5-Pro (please kindly check Tables 2,3 in our main paper).
> > >
> > > Table 1: SROCC (↑) and PLCC (↑) comparison on various sizes of training datasets. The Δ SROCC and Δ PLCC represent the performance improvement between the current samples and the -100 samples metrics.
> > > |training data size|SROCC(↑)|ΔSROCC|PLCC(↑)|ΔPLCC|
> > > |-----|-------|-------|-------|-------|
> > > | 2000 | 0.636 |     -     | 0.593 |     -     |
> > > | 2100 | 0.652 |  0.016    | 0.608 |  0.015    |
> > > | 2200 | 0.663 |  0.011    | 0.619 |  0.011    |
> > > | 2300 | 0.669 |  0.006    | 0.626 |  0.007    |
> > > | 2400 | 0.674 |  0.005    | 0.633 |  0.007    |
> > > | 2500 | 0.682 |  0.008    | 0.641 |  0.008    |
> > > | 2600 | 0.686 |  0.004    | 0.644 |  0.003    |
> > > | 2700 | 0.688 |  0.002    | 0.645 |  0.001    |
> > > | 2800 | 0.690 |  0.002    | 0.645 |  0.000    |
> > > | 2900 | 0.692 |  0.002    | 0.647 |  0.002    |
> > >
> > > We sincerely hope our response has addressed your concerns, and we would be happy to provide further clarification if needed.

---

### Official Review · Reviewer_1pGr · 2025-03-11

**Overall Recommendation:** 4

**Summary:**

This paper introduces a benchmark for evaluating T2V models. The authors sample 10 video generation models and employ five distinct annotators to perform CoT labeling on the outputs, providing both reasoning and scores. The resulting dataset is then used to fine-tune Qwen2-VL-7B. Across the seven evaluation dimensions and eight tested models, the proposed approach achieves performance surpassing all existing image-based LLMs and video-based LLMs. Notably, the evaluation framework presented in this work encompasses all essential aspects of high-level criteria, offering a comprehensive and well-rounded assessment methodology.

## update after rebuttal
The author show the generalization experiment compared with foundation models (i.e., GPT-4o, Gemini) and the performance is quite well, which addresses my concern. I raise my rate to 4.

**Claims And Evidence:**

Yes.

**Essential References Not Discussed:**

No.

**Experimental Designs Or Analyses:**

There is a generalization flaw in the comparison with other foundational VLMs and in the evaluation of video generation models.

1） Although the authors demonstrate superior performance across all seven evaluation dimensions compared to mainstream VLMs, this advantage may largely stem from the gap between real-world video data and synthetic video data. The high performance reported in this work could be attributed to a domain shift toward synthetic video data, which does not adequately validate the method's universality and generalization capabilities. A significant issue lies in the fact that the training data includes videos generated by the same models used for testing, resulting in an identical data domain. Consequently, the authors fail to provide evidence of the method's effectiveness on unseen video generation models, raising concerns about its applicability to broader or more diverse scenarios.

2）The fine-tuned model is trained on video data with a maximum duration of only 5 seconds, which limits its temporal understanding capability to extremely short time spans. However, the original foundational models are capable of comprehending videos at the minute level. This approach not only weakens the temporal understanding ability of the base model but also raises concerns about the scalability of the proposed method for future video generation models that may extend to durations of 30–60 seconds. The authors should provide an analysis of the model's performance on 10-second videos and compare it with the capabilities of the foundational model to demonstrate the generalization potential of the proposed method. Without such evidence, the applicability of the approach to longer videos remains questionable.

**Methods And Evaluation Criteria:**

1. The most critical aspect of evaluation is the model's generalization capability, especially after fine-tuning. In this paper, the authors use LoRA for fine-tuning, which, while cost-effective, inherently limits the model's generalization ability from the outset. This constraint introduces a potential bias and may lead to unfair evaluations for future video generation models, as the fine-tuned model might not adequately represent the broader capabilities of the original architecture or other competing approaches.
2. The decision to sample only three frames for constructing the CoT data during data collection is problematic. While this limitation arises from GPT-4o's constraint of only supporting image inputs, three frames may not sufficiently capture the temporal segments that annotators focus on. As a result, the derived CoT chains could become inconsistent or chaotic. Although the authors mention retaining only cases where both scores and reasoning are correct, this approach risks discarding challenging or fine-grained samples, which could otherwise provide valuable insights into model performance on more complex scenarios.

**Other Comments Or Suggestions:**

None.

**Other Strengths And Weaknesses:**

None.

**Questions For Authors:**

My primary focus is on the generalization performance of the method proposed in this paper, as well as the difficulty level of the benchmark it introduces. Below, I outline three experiments that I believe are necessary to thoroughly evaluate these aspects. If the authors can provide reasonable and satisfactory results for these experiments, I would be inclined to raise my score. Conversely, the absence of such results would lead me to maintain or lower my current evaluation.

1) A performance comparison with the foundation model in the case of longer videos is needed, which should include both models seen in the training data and unseen models, such as the 10-second version of CogVideo-1.5-5B or CogVideoX-2.

2) A performance comparison between video generation models not included in the training set and foundational models, such as CogVideoX , HunyuanVideo , and wanx , is necessary.

3) Since the training data only uses three frames processed by GPT-4o to construct the CoT, I believe that the retained samples are likely to be relatively coarse-grained and easy. I request the authors to showcase challenging samples from the test set and provide a performance comparison on these difficult samples relative to the foundational model.

4) For evaluation, reproducibility is very important, but it is also very challenging for VLMs. Please provide the mean and variance of the results for one dimension based on five repeated trials.

**Relation To Broader Scientific Literature:**

This paper provides a promising direction for more efficient and effective evaluation of video generation models, an increasingly critical challenge as these models continue to advance. Traditional evaluation algorithms struggle to comprehensively assess high-level semantic dimensions, which often require the use of VLMs to address. Additionally, there is currently a lack of robust automated evaluation methods for diverse assessment dimensions. By fine-tuning a foundational model to tackle this issue, the authors demonstrate preliminary feasibility, showcasing its potential to streamline and enhance the evaluation process. This paradigm is also applicable to video understanding tasks in MLLMs, highlighting its broader relevance and utility in addressing challenges at the intersection of video generation and multi-modal reasoning.

**Theoretical Claims:**

No theoretical claim.

---

> ### Author Rebuttal · Authors · 2025-03-30
>
> We sincerely appreciate your detailed and constructive feedback. We are grateful that you recognized the strengths of our work, including (1) "**comprehensive and well-rounded**" method, (2) "provides a **promising** direction", (3) and "highlighting its **broader relevance and utility** in addressing challenges at the intersection of video generation and multi-modal reasoning". Your comments and questions are highly valuable, and we would like to address your question as follows.
>
> **[Q1] LoRA fine-tuning.**
>
> **[A1]** Thank you for your insightful comments. We acknowledge that full fine-tuning can offer greater performance and generalization. However, due to computational resource constraints, we opted for LoRA fine-tuning, which enables efficient adaptation with fewer trainable parameters. We believe that fine-tuning should introduce task-specific adaptation while leveraging the video comprehension and reasoning capabilities of MLLMs for semantic-based evaluation. LoRA provides a sufficiently efficient and cost-effective solution in this context.
>
> **[Q2] Long AIGC videos.**
>
> **[A2]** Considering time and computational constraints, we first analyzed the T2V model benchmarking results and selected the 20 prompts with the lowest mean scores across all model-generated video evaluations in each dimension as “difficult benchmark”. Using an RTX 3090, we generated 81-frame, 8 FPS, 10-second videos with CogVideoX1.5 based on these prompts as long video data. Three annotators rated these videos on a scale of 1-5, and we then evaluated and compared the results using both our evaluation model and the base model, reporting their respective correlations with human annotations.(See https://imgur.com/sc2BMVa)
>
> **[Q3] More recent T2V models benchmarking.**
>
> **[A3]** We evaluated more advanced T2V models on the previously defined “difficult benchmark”. Kling 1.6 generated 5-second standard videos using the official API, while Hunyuan and Wan 2.1 generated 5-second videos via the SiliconCloud API. Three annotators rated these videos on a scale of 1-5, and we then assessed and compared their correlation with human annotations using both our evaluation model and the base model. The results are as follows.(See https://imgur.com/71BbB6A)
>
> **[Q4] Challenging samples.**
>
> **[A4]** Thank you for your professional and insightful suggestions. Further optimization of thought chains is indeed a promising direction for improvement. Our work serves as an initial exploration of thought chaining in video evaluation, showcasing the potential of MLLM reasoning in this domain. Future research can build upon our approach for further refinements. To address your concerns, we tested our model on a challenging sample set(10 longest prompts per dimension) and observed a significant improvement in correlation compared to the base model.
>
> |  | Avg | Quality | Aesthetic | Consistency | Alignment | Rationality | Safety | Creativity |
> | --- | --- | --- | --- | --- | --- | --- | --- | --- |
> | Base | 0.462/0.202/1.109 | 0.595/0.527/0.636 | 0.294/-0.135/0.8 | 0.261/-0.392/0.727 | 0.479/0.123/0.9 | 0.436/0.345/1.3 | 0.697/0.375/2.6 | 0.473/0.569/0.8 |
> | GRADEO | 0.733/0.673/0.296 | 0.889/0.853/0.273 | 0.864/0.764/0.1 | 1.0/1.0/0.0 | 0.612/0.395/0.4 | 0.503/0.405/0.5 | 0.758/0.673/0.200 | 0.509/0.623/0.600 |
>
> Below is an example along with a simple evaluation comparison between GRADEO and the Base Model:
>
> https://imgur.com/a/nCu81gv
>
> (Dimension: Rationality, Prompt: Water at 100°C, Human: 2)
>
> (Base Model: …The event depicted, water being poured into a glass, is a common-sense action that aligns with real-world expectations and logical consistency in common sense knowledge and reasoning…I would rate the rationality of this video as 5…)
>
> (GRADEO: …prompt involves water at 100°C, which is the boiling point of water…The splash is not sustained, and there is no visible steam rising, which is a critical aspect of boiling water…Rating Score: 2…)
>
> **[Q5] Model reproducibility.**
>
> **[A5]** Thank you for raising this important point. We conducted five repetitions on Rationality, which resulted in some variations in individual data points, as shown in the table. Given that **human evaluations also exhibit disagreement** and that our **scores are coarse-grained (integers from 1 to 5)**, we believe the model demonstrates sufficient stability.
>
> |  | exp1 | exp2 | exp3 | exp4 | exp5 | mean | variance |
> | --- | --- | --- | --- | --- | --- | --- | --- |
> | SROCC,PLCC,MAE | 0.606,0.515,0.56 | 0.613,0.55,0.56 | 0.635,0.541,0.54 | 0.585,0.492,0.56 | 0.634,0.562,0.54 | 0.615,0.532,0.552 | 3.5e-4,6.39e-4,9.6e-5 |
>
> ---
>
> Finally, we deeply appreciate the thoughtful questions and suggestions, which have greatly contributed to improving our work and will be incorporated into our revised manuscript. **We sincerely hope it sufficiently address your concerns and earn your recognition.**

---

> > ### Comment · Reviewer_1pGr · 2025-04-02
> >
> > Thank you very much for the author's supplementary experiments. However, I still have some questions regarding the implementation details of these supplementary experiments. What is the base model you are comparing against? If it is Qwen2-VL-7B, then I believe there might be a misunderstanding of my point. The generalization comparison I am referring to should be relative to all the foundational models you are comparing, not just one specific model. You should be comparing against models like GPT-4V or Gemini-1.5. When conducting benchmarks, the two most critical aspects are always generalization and accuracy. If current foundational models like GPT-4o already surpass your method in terms of generalization, then the entire evaluation system becomes meaningless.

---

> > > ### Author Response · Authors · 2025-04-07
> > >
> > > **To address the reviewer's concern, we further compare our method against the most advanced MLLMs, GPT-4o and Gemini 1.5 Pro, on both real-world datasets and SOTA video generation models, demonstrating the superior generalization capability of our approach.**
> > >
> > > **[A] Generalizability on Real-World Video and New T2V Models (Comparison with GPT-4o and Gemini 1.5 Pro)**
> > >
> > > - To better demonstrate the generalizability of our model on video scoring tasks, we constructed a real-world video dataset derived from Panda-70M and OpenVid-1M. Though our model was not trained on any real-world data, the video assessment capability learned by our GRADEO generalizes well to real video data. In fact, our model significantly outperforms SOTA models such as GPT-4o and Gemini 1.5 Pro (see Table 1).
> > >
> > > - As discussed in Rebuttal A2 and A3, we also evaluated our model on **unseen**, AI-generated videos produced by recent models including Kling-1.6, Hunyuan, and Wan 2.1-14B, none of which were included in our training set. Despite this, our model still achieves significant performance improvements, demonstrating its robust generalization (see Table 2).
> > >
> > > - While GPT-4o and Gemini 1.5 Pro are powerful, they are not specifically aligned with human video preferences. So they can not directly be used for video evaluation, highlighting the importance of our human-annotated data. Our approach shows better human alignment, offering a stronger path for T2V evaluation.
> > >
> > > **[B] Generalizability on Long Video(Comparison with GPT-4o and Gemini 1.5 Pro)**
> > >
> > > - As there are currently few AIGC datasets with 30s videos, we test on real-world 30s+ video. Our model achieves significantly better performance than GPT-4o and Gemini 1.5 Pro, demonstrating its superior capability to generalize to longer video (see Table 1).
> > >
> > > - Additionally, in Rebuttal A2, we evaluate our model on generated long videos by introducing 10s videos from CogVideoX (as the reviewer suggested). Our model also achieves the best, further supporting its generalizability to longer video (see Table 3).
> > >
> > > Table 1: Generalizability on real-world video data.
> > > ||Avg|Quality|Aesthetic|Consistency|Alignment|Rationality|Safety|Creativity|
> > > |---|---|---|---|---|---|---|---|---|
> > > |GPT-4o(5s)|.344/.311/1.143|.516/.489/1.000|.150/.105/1.360|.273/.234/1.240|.506/.481/.920|.134/.080/1.280|.460/.444/1.100|.371/.342/1.100|
> > > |Gemini-1.5-Pro(5s)|.411/.387/.980|.368/.319/1.120|.195/.156/1.220|.550/.536/.740|.575/.567/.860|.377/.330/1.040|.382/.351/.960|.431/.449/.920|
> > > |Qwen2-VL-7B (5s)|.361/.333/1.106|.378/.356/1.060|.331/.308/1.100|.380/.352/1.080|.491/.458/1.100|.462/.436/.920|.401/.352/1.100|.086/.072/1.380|
> > > |GRADEO(5s)|.727/.715/.489|.769/.742/.500|.730/.723/.480|.843/.831/.420|.786/.772/.420|.552/.533/.700|.686/.681/.440|.725/.721/.460|
> > > ||
> > > |GPT-4o(30s-60s)|.329/.293/1.063|.349/.306/1.100|.422/.397/.960|.174/.118/1.200|.276/.221/1.100|.251/.228/1.020|.425/.402/1.020|.407/.377/1.040|
> > > |Gemini-1.5-Pro(30s-60s)|.354/.310/1.011|.315/.282/.960|.489/.451/.820|.367/.329/.880|.252/.192/1.220|.241/.175/1.100|.340/.293/1.180|.475/.450/.920|
> > > |Qwen2-VL-7B (30s-60s)|.274/.231/1.191|.283/.265/1.220|.241/.152/1.260|.210/.170/1.220|.390/.372/1.020|.171/.127/1.280|.269/.215/1.260|.352/.316/1.080|
> > > |GRADEO(30s-60s)|.666/.644/.529|.681/.664/.540|.616/.595/.520|.730/.702/.340|.705/.682/.520|.564/.530/.620|.673/.650/.580|.695/.685/.580|
> > >
> > > Table 2:  Average score on new T2V models (Kling1.6, Hunyuan, Wan2.1-14B).
> > > ||Avg|Quality|Aesthetic|Consistency|Alignment|Rationality|Safety|Creativity|
> > > |---|---|---|---|---|---|---|---|---|
> > > |GPT-4o|.375/.267/.947|.412/.335/.767|.308/.190/.833|.281/.002/.833|.386/.300/.967|.420/.394/1.033|.429/.402/1.117|.384/.248/1.083|
> > > |Gemini-1.5-Pro|.384/.298/.895|.228/.101/1.083| .404/.247/.650|.269/.200/.783|.378/.278/1.033|.445/.391/.983|.444/.432/.967|.522/.441/.767|
> > > |Qwen2-VL-7B|.322/.191/1.033|.333/.247/.917|.350/.215/.933|.452/.201/.717|.405/.196/.783|.055/-.017/1.483|.395/.382/1.133|.263/.114/1.267|
> > > |GRADEO|.629/.529/.548|.701/.628/.483|.514/.426/.617|.534/.306/.667|.498/.359/.683|.662/.626/.500|.794/.751/.483|.699/.610/.400|
> > >
> > > Table 3: Generalizability on long CogVideoX data(10s).
> > > ||Avg|Quality|Aesthetic|Consistency|Alignment|Rationality|Safety|Creativity|
> > > |---|---|---|---|---|---|---|---|---|
> > > |GPT-4o|.261/.163/.893|.323/.207/.900|.216/.192/.800|.481/.347/.700|.315/.259/.700|.184/.103/1.050|.128/.018/1.250|.179/.013/.850|
> > > |Gemini-1.5-Pro|.384/.325/.779|.340/.228/.950|.352/.318/.700|.506/.582/.600|.315/.259/.700|.311/.218/.950|.274/.134/1.000|.590/.538/.550|
> > > |Qwen2-VL-7B|.363/.253/.850|.246/.212/1.000|.335/.266/.750|.512/.481/.550|.657/.543/.600|.337/.360/.950|.219/-.069/1.100|.236/-.020/1.000|
> > > |GRADEO|.602/.496/.586|.482/.297/.650|.618/.495/.550|.691/.730/.500|.638/.483/.500|.776/.747/.550|.550/.400/.650|.462/.323/.700|
> > >
> > > We have made every effort to address your concerns and hope the improvements demonstrate the strengths of our approach. We kindly ask for your reconsideration.

---

### Official Review · Reviewer_hvju · 2025-03-13

**Overall Recommendation:** 4

**Summary:**

This paper introduces GRADEO, a novel approach for evaluating text-to-video (T2V) generation models using human-like multi-step reasoning. The authors identify key limitations of existing evaluation methods, which often lack high-level semantic understanding and reasoning capabilities, making them inadequate for comprehensive video assessment. To address this gap, they created GRADEO-Instruct, a dataset containing 3.3k videos with human annotations across seven evaluation dimensions (Quality, Aesthetic, Consistency, Alignment, Rationality, Safety, and Creativity). The trained GRADEO and employed a four-step reasoning process: Overview, Description, Analysis, and Assessment, which mimics human evaluation practices. Experiments proved that it's effective in both correlation with human preference and pairwise comparison accuracy.

## update after rebuttal
The authors' rebuttal resolved most of my concerns and I think it's a good paper to accept. However, there is still a limitation of the efficiency of the model, as said by the authors "takes 30 seconds and generates about 400 tokens per video". Since these kinds of scoring models will eventually be used for cases like RL for diffusion models, BoN selection, 30 seconds per generation seems not quite acceptable. However, this is not a big concern as there are many acceleration frameworks like vllm, sglan and I encourage the authors to try integrating the model into them. Overall, I think it's a good paper to be accepted

**Claims And Evidence:**

yes

**Essential References Not Discussed:**

N/A

**Experimental Designs Or Analyses:**

The experimental designs are generally sound. The comparison with baseline methods includes both automated metrics and LLM-based approaches. And the pairwise evaluation on existing datasets (T2VQA-DB, GenAI-Bench-Video, TVGE) helps validate the model's effectiveness.

However, the paper mentions filtering videos with "unsuitable dynamics" but doesn't fully define what criteria were used, which could introduce selection bias.

**Methods And Evaluation Criteria:**

Yes. The introduce of CoT in Video Evaluation is a great attempt. Also the definition of 7 evaluation aspects make sense. The pairwise benchmarks used for selection also makes sense.

**Other Comments Or Suggestions:**

1. A discussion on the cost-efficiency tradeoff compared to other evaluation approaches would be valuable. Like how many CoT token will GRADEO usually output.

**Other Strengths And Weaknesses:**

### Strengths:
- The Chain-of-Thought reasoning process creates interpretable assessments beyond simple scores.
- The qualitative examples effectively demonstrate the issues in current T2V models.

### Weaknesses:

1. The dataset size (3.3k videos) is relatively small compared to some other benchmark datasets.
2. The seven evaluation dimensions may have some overlap or correlation, which isn't deeply analyzed.
3. The video generations sources, i.e. the t2v models, are limited to Sora, VideoCrafter-1, VideoCrafter-2, Latte, LaVie, and ZeroScope-576w. From my experience, these models are usually not good enough and nowadays there are many more powerful t2v models like wanxiang, stablevideodiffusion, etc. Whether GRADEO can generalize to these model's outputs are unknown

**Questions For Authors:**

1. How did you determine the four-step reasoning process (Overview, Description, Analysis, Assessment)? Did you experiment with other reasoning structures, and if so, how did they compare?
2. The VideoScore has less competitive results on Table 2 (your developed benchmark) compared to Table 3 (existing benchmarks). Are there any biases in the results of Table 2? Do you have any insights on this?

**Relation To Broader Scientific Literature:**

- There are already previous works like VideoScore that develop specific models for text-to-video evaluation. However, none of these works have ever employed CoT reasoning during the model training. This paper's most significant contribution is the CoT based evaluation dataset and proved that CoT-based evaluation works well.

**Theoretical Claims:**

There is not theoretical claim in the paper.

---

> ### Author Rebuttal · Authors · 2025-03-30
>
> We sincerely thank you for your time and appreciate your valuable comments. We are motivated to see that the reviewer finds our work (1) **the novelty and validity** of introducing CoT, (2) **the comprehensiveness and rationalization** of dimension definitions and experimental setups, (3) CoT reasoning process creates **interpretable assessments beyond simple scores**. Then we will explain your concerns point by point.
>
> **[Q1] Definition of inappropriate dynamics.**
>
> **[A1]** We filtered video data with "inappropriate dynamics" based on our observations during data collection. In AIGC-video datasets like T2VQA-DB[1], some videos, generated by earlier models, showed abrupt frame transitions with poor dynamic consistency, while others were nearly static.
>
> Human evaluators easily identified these low-quality videos, which could affect model evaluation accuracy. To ensure dataset quality, we applied SSIM and FLOW scores (see Appendix A.2), with thresholds based on human visual perception.
>
> **[Q2] Concerns about the size of the dataset.**
>
> **[A2]** To minimize individual evaluator bias, we have five evaluators assess each video and exclude data points where there are significant discrepancies in evaluations. Though cost limits scale, our approach is still scalable and generalizable, and we hope it will inspire further advancements in video generation.
>
> Additionally, previous AIGC quality assessment datasets are relatively small: T2VQA-DB [1] contains 10,000 videos, MQT [2] has 10,005, FETV [3] includes 2,476, and LGVQ [4] has 2,808. While previous datasets mainly rely on human ratings, our dataset includes both ratings and annotated assessment rationales, requiring deeper analysis and increasing costs. These datasets only provide human subjective ratings, which are not suitable for training interpretable models.
>
> **[Q3] Overlap or correlation among seven dimensions.**
>
> **[A3]** Human assessment is the gold standard, but fully isolating evaluation dimensions remains challenging. To address this, we define more granular sub-dimensions for each assessment dimension, aiming to provide a more comprehensive and mutually independent evaluation framework.
>
> **[Q4] Limitations of video generation sources.**
>
> **[A4]** To address your concerns, we benchmarked recently introduced T2V models using our model(See https://imgur.com/wzQubJK). Considering the time and arithmetic constraints, we first counted the data of the T2V models from previous benchmarking papers and took out the 20 prompts with the lowest average scores of all model-generated video evaluations out of the 100 prompts for each dimension as the difficult benchmarks.
>
> **[Q5] The cost-efficiency tradeoff.**
>
> **[A5]** On average, GRADEO takes 30 seconds and generates about 400 tokens per video. While more computationally expensive than automatic metrics, these metrics fail to capture high-level semantics and lack explainability.
>
> **[Q6] 4 steps reasoning process.**
>
> **[A6]** First, we identified the need for a two-step process: "Analysis" and "Assessment", where score is the goal and analysis extends human reasoning, aligning the process more closely with human thought for more accurate evaluations. With the introduction of the "Description" step, the model’s outputs are more grounded in video content analysis. The sub-dimensions of reasoning and assessment vary across evaluation dimensions, and the "Overview" step ensures reasoning stays within the intended dimension. We conducted ablation experiments to highlight the importance of the four-step reasoning framework.
>
> **[Q7] The videoscore results issue.**
>
> **[A7]** This is an interesting question, and we believe several factors may contribute: (1) Pairwise comparative assessment focuses on relative ranking rather than absolute scoring. For example, given a pair of videos with human scores of <video 1>: 4 and <video 2>: 5, the model only needs to rank video 1 lower than video 2 to align with human judgment. This results in outputs like (1,2), (2,3), or (3,4), ensuring consistency in pairwise comparison but not necessarily in absolute scoring. (2) VideoScore produces scores in the range of 1-4, and we applied linear mapping and rounding to align them with our assessment scale, which may have introduced additional inconsistencies.
>
> [1] Subjective-aligned dataset and metric for text-to-video quality assessment. ACM-MM 2024.
>
> [2] Measuring the quality of text-to-video model outputs: Metrics and dataset.
>
> [3] FETV: A Benchmark for Fine-Grained Evaluation of Open-Domain Text-to-Video Generation. NeurIPS 2023
>
> [4] Benchmarking Multi-dimensional AIGC Video Quality Assessment: A Dataset and Unified Model.
>
> ---
>
> Finally, we deeply appreciate the thoughtful questions and suggestions, which have greatly contributed to improving our work and will be incorporated into our revised manuscript. **We sincerely hope it sufficiently address your concerns and earn your recognition.**

---

### Official Review · Reviewer_x7z3 · 2025-03-14

**Overall Recommendation:** 3

**Summary:**

This paper addresses the challenge of evaluating video generation models by introducing GRADEO, a novel video evaluation model designed to provide explainable scores and assessments through multi-step reasoning. The authors curate GRADEO-Instruct, a multi-dimensional dataset with 3.3k videos and 16k human annotations, enabling the model to better align with human evaluations. Experiments demonstrate that GRADEO outperforms existing automated metrics in capturing high-level semantic understanding and reasoning, revealing limitations in current video generation models to align with human reasoning and complex real-world scenarios.

**Claims And Evidence:**

Yes.

**Essential References Not Discussed:**

The preprint Evaluation Agent [1] is a similar evaluation protocol that also adopts an agent-based assessment strategy. I suggest the authors have a brief discussion towards the discrepancy against this paper.


[1] Zhang, Fan, et al. "Evaluation Agent: Efficient and Promptable Evaluation Framework for Visual Generative Models." arXiv preprint arXiv:2412.09645 (2024).

**Experimental Designs Or Analyses:**

The authors conduct two parts of experiments to verify the human alignment and evaluate current frontier T2V models, respectively.

1. For the human alignment experiments, the authors report the SROCC, PLCC, and mean absolute error and compare the proposed metric with several MLLMs. The proposed method outperforms all baseline methods in all metrics, yet I believe such results are not surprising since these baseline MLLMs are not specially designed for T2V evaluations, nor widely adopted as T2V evaluation metrics by previous methods. For the absolute value, the Spearman correlation coefficients of all dimensions fell into a threshold of 0.6-0.8, which I believe cannot be recognized as a strong correlation considering that the aesthetic, quality, and consistency correlation scores of VBench are more than 0.90. Therefore, compared with other T2V evaluation protocols, the proposed method seems not to show strong human preference correlations.

2. The authors select Qwen2-VL-7B model as the evaluation agent. Since there are some strong alternatives such as InternVL2.5 and Gemini, what will these models perform when using a similar instruction-tuning strategy on open-source MLLMs or use a few-shot CoT prompt for proprietary models? Will these models achieve higher human correlation scores?

3. For the benchmarking results, I found the score distribution in each dimension is very narrow and typically within 10 points, indicating the proposed benchmark may not effectively distinguish between models with significantly different performance levels. I suggest the authors conduct additional experiments on two models that perform significantly differently, such as Sora vs. some early-stage models to show whether the evaluation agent could give an extreme high or low scores.

4. The benchmarking is not sufficient. The authors should evaluate more recent models such as Hunyuan-Videos, StepVideo-T2V, Sora, Kling, etc. Results of current models are too similar to give some valuable insights.

**Methods And Evaluation Criteria:**

1. The authors select 7 aspects for evaluation. What is the rationale of choosing these dimensions? Are these dimensions suitable and enough to comprehensively evaluate T2V models? For example, how to evaluate whether objects in a video adhere to physical laws? how to evaluate whether the text, color, and font generated in the video according to the user's prompt is correct?

2. Descriptions on some dimensions are ambiguous. For example, how to assess the creativity and diversity of T2V models when the prompts for evaluation are pre-designed and fixed? The tested models may generate more diverse results based on more detailed, varied, and fine-grained prompts.

3. The capabilities of T2V models in generating lengths and resolution are different. As the authors claimed in Table 9, tested models generate videos in different durations, fps, and resolutions. How to guarantee the evaluation fairness when adopting a different experimental configuration? Besides, for videos that can generate videos in a significantly longer duration (such as videos more than 30 seconds) or much higher resolution, can the proposed benchmark highlight these aspects on one or more dimensions? More clarifications are needed.

4. The multi-step reasoning process is over-claimed. I believe the so-called multi-step reasoning process is actually a single-step fine-grained perception + single-step reasoning since I do not find any multi-step reasoning chains either from the CoT prompt templates and the model responses.

**Other Comments Or Suggestions:**

N/A

**Other Strengths And Weaknesses:**

N/A.

**Questions For Authors:**

1. What is the evaluation performance of Qwen2-VL-7B without instruction tuning? The authors should report the correlation in Table 2 to probe the effectiveness of the instruction tuning.

2. How does the proposed evaluation protocol prevent the negative influence of the LLM hallucinations? Though the authors explicitly discuss the issues in Appendix E, I believe some possible measures could be adopted to minimize the harm as much as possible, such as using prompts with clear and straightforward instructions, or using a self-reflection manner to automatically fix any potential mistakes.

3. As the authors claim in the main manuscript, the evaluation process requires high reasoning capability, how about selecting MLLMs that are trained with reasoning abilities, such as llava-cot, internvl-mpo, or video-of-thought? On the contrary, to what extent does the deficiency in reasoning capabilities of existing MLLMs affect the overall evaluation accuracy?

**Relation To Broader Scientific Literature:**

The key contribution could benefit the evaluation of text-to-video generation models.

**Theoretical Claims:**

The paper does not propose any new theoretical claim.

---

> ### Author Rebuttal · Authors · 2025-03-30
>
> We thank the reviewer for their time and appreciate that they valued the thorough evaluation. We would like to address your question as follows. Sorry for the space constraints, the table is provided via an anonymous link: https://imgur.com/a/VzqdmYY.
>
> **[Q1] Experimental setup.**
>
> **[A1]** We would like to clarify and analyze the experimental setup through the following points.
>
> (1) Dimensions setup: Our seven evaluation dimensions are designed to comprehensively assess both low-level perceptual quality and high-level semantic consistency in AI-generated videos. Our framework remains extendable for future enhancements.
>
> (2) Evaluation fairness: Many studies [1,2,3,4] lack uniform preprocessing for evaluation AIGC videos (EvalCrafter only adds watermarks for fairness). Preprocessing can alter video features, affecting model evaluation. Enforcing a uniform frame rate may distort motion quality in models optimized for high-frame-rate generation [5,6]. Segmenting long videos for comparison with short-video models disrupts temporal coherence and narrative fluency. To ensure fairness, we only remove small, corner-positioned watermarks, preserving video integrity.
>
> **[Q2] Human correlation.**
>
> **[A2]** We would like to clarify the following points:
>
> (1) It is not fair to directly compare VBench's correlations with those in Table 2. In Section 3.3 of VBench [1], relative comparisons were conducted using four T2V models, whereas our evaluation is based on an absolute 1-5 scoring scale. Aligning absolute scores with human assessments is inherently more challenging than aligning relative rankings.
>
> (2) Other T2V evaluation methods[4,7,8,9,10] also exhibit limited correlation with human assessments, likely due to the inherent variability in human judgments and the complexity of the evaluation task.
>
> (3) The MLLM-as-a-Judge [11] study further supports the difficulty of absolute scoring tasks for MLLMs compared to pairwise comparisons.
>
> (4) We conducted LoRA fine-tuning on InternVL2.5-4B for computational resource constraints(See anonymous link).
>
> **[Q3] Narrow score gap and more recent T2V models benchmarking.**
>
> **[A3]** This phenomenon is expected. Achieving a baseline quality score is easy, but high scores are harder. Advanced T2V models like Hunyuan and Wanxiang now focus on fine details. Thank you for your suggestions.
>
> We benchmarked more T2V models (hunyuan, wan2.1, kling1.6, cogx1.5, See anonymous link). Due to time and computational constraints, we identified "difficult benchmark" by selecting the 20 lowest-scoring prompts from previous benchmarking papers across 100 prompts per dimension.
>
> **[Q4] Evaluation Agent.**
>
> **[A4]** Thank you for highlighting this related work. (1) Evaluation Agent(contemporaneous with our work) introduces a dynamic, agent-based framework with hierarchical assessments but raises fairness concerns in evaluating diverse models. It lacks human annotations but offers flexibility and efficiency. (2) Our work develops MLLM-based models focused on reasoning with a curated human assessment dataset. Through instruction fine-tuning, our model demonstrates novelty and scalability in advanced semantic assessment.
>
> **[Q5] Performance of base model.**
>
> **[A5]** We will include the correlation between the base model and human evaluations in Table 2 of the revised version(See anonymous link).
>
> **[Q6] MLLM hallucinations.**
>
> **[A6]** We categorize hallucinations into two types:
>
> (1) Detachment from video content or inclusion of nonexistent elements: We address this with "Description" step, ensuring assessments stay grounded in actual video content, reducing hallucinations that affect reasoning and scoring.
>
> (2) Lack of focus on the intended assessment dimension: We mitigate this with "Overview" step, structuring the evaluation process and emphasizing sub-dimensions to maintain focus.
>
> Ablation experiments show that removing these steps leads to hallucinations and a quantitative drop in consistency with human evaluations.
>
> **[Q7] Four steps evaluation and Reasoning MLLMs.**
>
> **[A7]** We agree that an MLLM trained on reasoning datasets could improve evaluation accuracy. Future work could enhance accuracy by combining MLLMs with stronger reasoning and video comprehension capabilities alongside human evaluation datasets.
>
> [1] VBench. CVPR 2024
>
> [2] Towards A Better Metric for Text-to-Video Generation.
>
> [3] AIGCBench. TBench 2024
>
> [4] EvalCrafter. CVPR 2024
>
> [5] ZeroSmooth: Training-free Diffuser Adaptation for High Frame Rate Video Generation.
>
> [6] StreamingT2V. CVPR 2025
>
> [7] FETV: A Benchmark for Fine-Grained Evaluation of Open-Domain Text-to-Video Generation. NeurIPS 2023
>
> [8] Subjective-aligned dataset and metric for text-to-video quality assessment. ACM-MM 2024.
>
> [9] VideoScore. EMNLP 2024
>
> [10] Evaluating Text-to-Visual Generation with Image-to-Text Generation. ECCV 2024
>
> [11] MLLM-as-a-Judge: Assessing Multimodal LLM-as-a-Judge with Vision-Language Benchmark. ICML 2024

---

> > ### Comment · Reviewer_x7z3 · 2025-04-08
> >
> > I thank the authors for their detailed response. After reading through the authors' rebuttal and other expert reviewers' comments, most of my original concerns are addressed, and I raise my rating to weak acceptance.

---

> > > ### Author Response · Authors · 2025-04-08
> > >
> > > Thank you very much for raising your score! We're glad that our response has addressed your concerns. We truly appreciate your valuable comments, which will continue to guide us in improving our work.

---

### Decision · Program_Chairs · 2025-05-01

**Decision:**

Accept (poster)

**Comment:**

The paper initially received divergent reviews—two positive and two negative—but after a successful rebuttal, all four reviewers reached a positive consensus, with final scores of two accepts and two weak accepts.

The paper presents a novel T2V evaluation framework along with an instruction tuning benchmark. While initial concerns were raised regarding LLM hallucination, dataset scale, video generation sources, and generalization performance, the authors addressed these issues convincingly during the rebuttal.

In line with the reviewers, the AC finds that the strengths of the work outweigh the weaknesses and recommends acceptance.